# Coupled nitrification and N₂ gas production as a cryptic process in oxic riverbeds

Liao Ouyang[1,2], Bo Thamdrup [3] & Mark Trimmer [2✉]

The coupling between nitrification and N₂ gas production to recycle ammonia back to the atmosphere is a key step in the nitrogen cycle that has been researched widely. An assumption for such research is that the products of nitrification (nitrite or nitrate) mix freely in the environment before reduction to N₂ gas. Here we show, in oxic riverbeds, that the pattern of N₂ gas production from ammonia deviates by ~3- to 16-fold from that predicted for denitrification or anammox involving nitrite or nitrate as free porewater intermediates. Rather, the patterns match that for a coupling through a cryptic pool, isolated from the porewater. A cryptic pool challenges our understanding of a key step in the nitrogen cycle and masks our ability to distinguish between sources of N₂ gas that 20 years' research has sought to identify. Our reasoning suggests a new pathway or a new type of coupling between known pathways in the nitrogen cycle.

[1] College of Physics and Optoelectronic Engineering, Shenzhen University, Shenzhen 518060, China. [2] School of Biological and Chemical Sciences, Queen Mary University of London, London E1 4NS, UK. [3] Nordcee, Department of Biology, University of Southern Denmark, 5230 Odense M, Denmark. ✉email: m.trimmer@qmul.ac.uk

Nitrogen is a key bio-element for life on Earth, integral to proteins and the very DNA that tells life what to do. A vast reservoir of nitrogen resides in the atmosphere as $N_2$ gas, unavailable to the majority of life until being fixed by either biological or anthropogenic nitrogen fixation. Life's organically-bound nitrogen in turn decays to ammonia following excretion or death. To complete the cycle, first nitrogen must be oxidised to nitrite or nitrate which can then be reduced back to atmospheric $N_2$ gas. This process of ammonia oxidation—known as nitrification—typically occurs in two stages carried out by specialised aerobic chemoautotrophic ammonia- and nitrite-oxidising microbes, for example, in soils, sediments, freshwater, or marine ecosystems (Eqs. 1 and 2, respectively):

$$2NH_4^+ + 3O_2 \rightarrow 2NO_2^- + 2H_2O + 4H^+ \qquad \Delta G^{o\prime} = -270 \text{ kJ (per } NH_4^+) \quad (1)$$

$$2NO_2^- + O_2 \rightarrow 2NO_3^- \qquad \Delta G^{o\prime} = -79 \text{ kJ (per } NO_2^-) \quad (2)$$

Nitrite and nitrate can then be reduced to $N_2$ gas either alone, in a phylogenetically widespread form of microbial anaerobic respiration termed denitrification[1] (Eq. 3a, b) or, in combination with ammonia, in a phylogenetically narrow respiratory pathway termed anaerobic ammonia oxidation, namely anammox[2] (Eq. 4).

$$2NO_3^- + 10e^- + 12H^+ \rightarrow N_2 + 6H_2O \qquad \Delta G^{o\prime} = -360 \text{ kJ (per } NO_3^-) \quad (3a)$$

$$2NO_2^- + 6e^- + 8H^+ \rightarrow N_2 + 4H_2O \qquad \Delta G^{o\prime} = -282 \text{ kJ (per } NO_2^-) \quad (3b)$$

$$NH_4^+ + NO_2^- \rightarrow N_2 + 2H_2O \qquad \Delta G^{o\prime} = -358 \text{ kJ} \quad (4)$$

In addition, smaller amounts of N can be returned to the atmosphere as nitrous oxide ($N_2O$) but we do not consider those further here[3–5]. Combinations of Eqs. (1) to (4) recycle ammonia back into atmospheric $N_2$ gas and this coupling between aerobic nitrification and anaerobic $N_2$ gas production is a key concept in the nitrogen cycle, controlling ecosystem production and the abundance of life on Earth[6,7].

Besides the now accepted reactions described in Eqs. (1) to (4), Broda's original thermodynamic predictions that drove the quest for anammox[8,9] also included the potential for complete aerobic ammonia oxidation to $N_2$ gas—that, to the best of our knowledge—has yet to be observed in nature:

$$4NH_4^+ + 3O_2 \rightarrow 2N_2 + 6H_2O + 4H^+ \qquad \Delta G^{o\prime} = -316 \text{ kJ (per } NH_4^+) \quad (5)$$

In estuarine or coastal sea sediments, combinations of recognised aerobic and anaerobic metabolisms (Eqs. 1 to 4) buffer the flux of terrestrial nitrogen out to sea and are considered to be physically divided between the oxic and anoxic sediment layers—albeit by only a few tenths of millimetres[10]. In rivers, nitrite and nitrate borne from aerobic nitrification (Eqs. 1 and 2), in either the surrounding catchment soils or the riverbed itself, can be transported over large distances (1–100 km) before some 47 Tg N per year is removed from the fluvial network as $N_2$ gas[11–13]. Regardless of the setting, the important point to appreciate here is that the products of aerobic nitrification (e.g., nitrate and nitrite) are assumed to be free to mix with any existing nitrate and nitrite in the surrounding porewater before they are subsequently metabolised, anaerobically, to $N_2$ gas. That is, there is—in effect—only one pool of nitrate and nitrite awaiting reduction to $N_2$ gas regardless of their origins. Indeed, this concept of free mixing between substrates lies at the very heart of the common $^{15}N$ isotope pairing techniques used to disentangle and quantify the cycling of nitrogen in sediments that are major sources of $N_2$ gas on Earth[11,14,15].

Most research into the coupling between aerobic nitrification and anaerobic $N_2$ gas production in sediments has studied the two separately using either oxic or anoxic incubations, respectively[16], but now work including oxygen is increasing[17]. Previously we demonstrated[18] that oxic (~30% to 100% of air-saturation for oxygen) gravel and sandy riverbed sediments harbour a coupling between aerobic nitrification and, seemingly, anaerobic $N_2$ gas production with that production being attributed to a combination of denitrification and anammox[18]. We now show that the pattern of $N_2$ gas production from ammonia in these oxic riverbeds violates the prevailing concept that coupled nitrification and $N_2$ gas production is a two-step process with free nitrite or nitrate as intermediates. Not only does this challenge our understanding of a key coupling in the nitrogen cycle but it also masks our ability to distinguish between denitrification and anammox as sources of $N_2$ gas. Indeed, it may actually suggest a new pathway or at least a new type of coupling between known pathways in the nitrogen cycle.

## Results and discussion

**$N_2$ gas production is independent from porewater nitrite or nitrate.** Following on from our original work[18] on nitrification and putative anaerobic $N_2$ gas production in oxic riverbeds, we wanted to explore further how these two processes are coupled. We began by collecting sediment from four rivers—two each of predominantly gravel and sand and then extended our sampling to a total of twelve rivers (Supplementary Figure 1 and Supplementary Table 1). We added $^{15}N$-ammonia to oxic sediment microcosms (see Methods) to trace the coupling between nitrification and $N_2$ gas production both with and without the inhibitor of aerobic nitrification, allylthiourea[19] (~80 μM ATU in the porewater, Treatments 1 & 2, Table 1 and Methods) that does not inhibit denitrification or anammox[2,20]. As before[18], we measured

**Table 1 Summary of total $^{15}N$-$N_2$ production in oxic incubations with $^{15}NH_4^+$ or $^{15}NO_2^-$. Mixed-effects models were used to estimate overall rates of total $^{15}N$-$N_2$ production for the incubations in Fig. 1a. Treatments 1 to 6 were applied to sediments from the first set of 4 rivers, and then just treatments 1 and 2 for the subsequent set of 12 rivers. Model fitting was carried out in the lme4 package in R[45] and rate estimates, standard errors (s.e.) and 95% confidence intervals derived using emtrends from the emmeans package (see Methods). Significant production (bold) of $^{15}N$-$N_2$ was only measured with treatments 1 and 3.**

| Code, Treatment | Rivers (replicates) | Total $^{15}N$-$N_2$ (nmol N g$^{-1}$ h$^{-1}$) | s.e. | Lower 95% C.I. | Upper 95% C.I. |
|---|---|---|---|---|---|
| **1**, $^{15}NH_4^+$ + ATU | 4 (5) | 0.110 | 0.337 | −0.667 | 0.886 |
| **2**, $^{15}NH_4^+$ | 4 (5) | **1.855** | 0.326 | 1.078 | 2.631 |
| **3**, $^{15}NH_4^+$ + $^{14}NO_2^-$ + ATU | 4 (5) | 0.152 | 0.337 | −0.625 | 0.929 |
| **4**, $^{15}NH_4^+$ + $^{14}NO_2^-$ | 4 (5) | **1.941** | 0.326 | 1.165 | 2.717 |
| **5**, $^{14}NH_4^+$ + $^{15}NO_2^-$ + ATU | 4 (5) | 0.314 | 0.326 | −0.462 | 1.091 |
| **6**, $^{14}NH_4^+$ + $^{15}NO_2^-$ | 4 (5) | 0.279 | 0.326 | −0.497 | 1.055 |
| **1**, $^{15}NH_4^+$ + ATU | 12 (5) | 0.129 | 0.178 | −0.249 | 0.506 |
| **2**, $^{15}NH_4^+$ | 12 (5) | **1.465** | 0.176 | 1.091 | 1.839 |

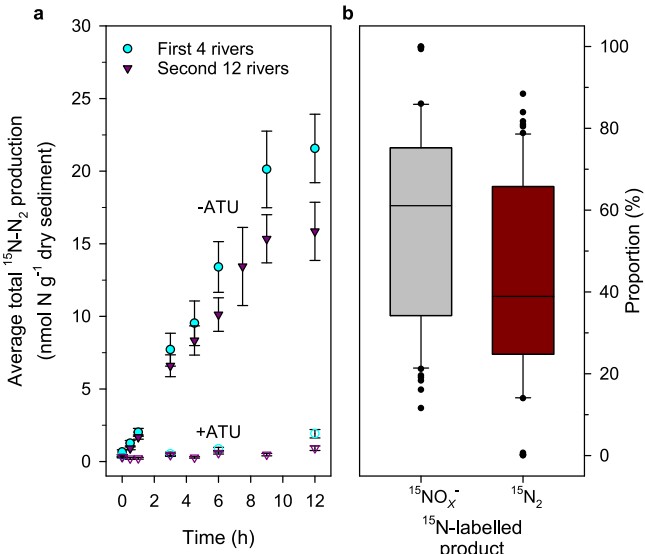

**Fig. 1 Oxic incubations with $^{15}$N-ammonia tracer produce both $^{15}$N$_2$ gas and $^{15}$NO$_x^-$. a** Overall average production of total $^{15}$N-N$_2$ (i.e., $^{29}$N$_2$ and $^{30}$N$_2$) over time in the presence or absence of the inhibitor of ammonia mono-oxygenase, allylthiourea (ATU). The first 4 rivers (cyan circles, $n = 40$, 4 rivers x 5 replicates x 2 treatments at each time point, ± 1 s.e.) and the follow-up across 12 rivers (purple triangles, $n = 60$, 12 rivers x 5 replicates at each time point, ± 1 s.e.); open coloured symbols are the same plus ATU (see Table 1). **b** Proportions of oxidised $^{15}$N-ammonia tracer from **a**, recovered as either $^{15}$NO$_x^-$ or $^{15}$N$_2$ across the 12 rivers ($n = 60$ as for **a**). Upper and lower box boundaries are 75th and 25th percentiles, respectively, upper and lower whiskers are 90th and 10th percentiles, respectively, the extreme outliers the maxima and minima and the horizontal line the centre, median value.

the immediate production of $^{15}$N-N$_2$-gas that was stopped by inhibiting the first step (Eq. 1) of aerobic $^{15}$N-ammonia oxidation with ATU (Fig. 1a, Table 1). The coupling between aerobic ammonia oxidation and N$_2$ gas production was clearly strong, however it was not complete. For example, across the twelve rivers, approximately 60% (Fig. 1b) of the oxidised $^{15}$N-ammonia tracer was recovered from the porewater as $^{15}$NO$_x^-$, i.e., as either $^{15}$N-nitrite (Eq. 1) or the final product of nitrification, $^{15}$N-nitrate (Eq. 2) e.g., $^{15}$NO$_x^-$ is the sum of $^{15}$NO$_2^-$ and $^{15}$NO$_3^-$.

The presence of $^{15}$N-ammonia and $^{15}$N-NO$_x^-$ together in the porewater generates two $^{15}$N-labelled substrate pools. The fraction of the pool labelled with $^{15}$N is termed $F_A$ for ammonia (NH$_3$) and $F_N$ for NO$_x^-$ (Eqs. 10 and 11 in Methods). Theoretically, combinations of Eqs. (1) to (4) can draw on these two substrate pools ($F_A$ and $F_N$) to produce both the single-$^{15}$N-labelled, $^{29}$N$_2$ gas (e.g., $^{14}$N, $^{15}$N) and the double-$^{15}$N-labelled, $^{30}$N$_2$ gas (e.g., $^{15}$N, $^{15}$N) which we illustrate schematically in Fig. 2a. Note that denitrification can draw on NO$_x^-$ as either NO$_2^-$ or NO$_3^-$ but anammox is solely fuelled by NO$_2^-$. The published and accepted mathematical framework[21] (See derivation of equations in Supplementary Note 1) tells us that the fraction of $^{15}$N-labelling in each of the substrate pools ($F_A$ and $F_N$) must influence the ratio of $^{29}$N$_2$ to $^{30}$N$_2$ (here termed $R$) and the overall fraction of $^{15}$N in the N$_2$ gas produced e.g., the overall blend of $^{28}$N$_2$, $^{29}$N$_2$ and $^{30}$N$_2$ (here termed $F_{N2}$)[21,22]. While complex, the accepted framework also tells us that so long as we know what fraction of each component part ($F_A$, $F_N$ and $F_{N2}$) is labelled with $^{15}$N, then we can still calculate how the N$_2$ gas is produced e.g., by anammox or denitrification and understand the nature of this key coupling in the nitrogen cycle[21,22].

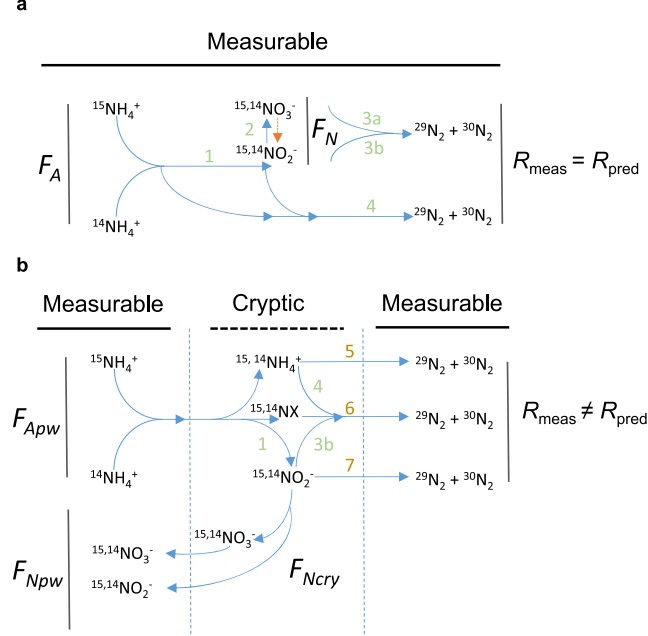

**Fig. 2 Accepted and proposed cryptic couplings in oxic N cycling.**
**a** $^{15}$NH$_4^+$ tracer is added to oxic sediments to mix with $^{14}$NH$_4^+$ in the porewater, with the fraction of $^{15}$N labelling known as $F_A$. Through reactions 1 and 2, $^{14}$NH$_4^+$ and $^{15}$NH$_4^+$ are oxidised aerobically to $^{14,15}$NO$_2^-$ and $^{14,15}$NO$_3^-$ to generate a $^{14,15}$NO$_x^-$ pool with $^{15}$N labelling known as $F_N$. NO$_3^-$ and/or NO$_2^-$ can be denitrified to N$_2$ gas (reactions 3a, 3b), or NO$_2^-$ can oxidise NH$_4^+$ anaerobically through anammox to N$_2$ gas (reaction 4). Regardless of the precise setting and combination of reactions, all substrates and products are free to mix and the measured ratio of $^{29}$N$_2$ to $^{30}$N$_2$ produced ($R$) can be predicted from the measured $^{15}$N labelling in the porewater. The downwards pointing orange arrow indicates NO$_3^-$ respiration to NO$_2^-$ that we do not consider further here. **b** In contrast, our measured values for $R$ cannot be predicted using the measured fraction of $^{15}$N labelling in the porewater ($F_A$ and $F_N$) and known combinations of reactions 1 to 4 but can only be approximated assuming a cryptic element ($F_{Ncry}$). A cryptic element could be a hidden substrate pool (6, novel or known) or novel parts of existing processes (7, e.g., complete nitrifier-denitrification beyond N$_2$O to N$_2$) and/or a completely new pathway (reaction 5 e.g., complete aerobic ammonia oxidation to N$_2$) or cryptic combinations of known pathways after partial aerobic ammonia oxidation to nitrite (reactions, 1, 3b, 4).

We tested the validity of this accepted mathematical framework by changing the fraction of porewater NO$_x^-$ labelled with $^{15}$N ($F_N$) and looking for how this influenced the ratio of $^{29}$N$_2$ to $^{30}$N$_2$ produced ($R$). First we directly decreased $F_N$ by adding $^{14}$N-nitrite to dilute the $^{15}$N-nitrite accumulating in the porewater from the oxidation of $^{15}$N-ammonia (Treatments 3 and 4, Table 1). Surprisingly, diluting $F_N$ had no discernible effect on the values for $R$ produced in the two sets of incubations (Fig. 3b. 2.32, 95% CI 2.01 to 2.64 versus 2.43, 95% CI 2.12 to 2.74, see Table 2 and Supplementary Table 2 for $^{29}$N$_2$ and $^{30}$N$_2$ production). We then repeated our incubations with just $^{15}$NH$_4^+$ (with and without ATU, Treatments 1 and 2) across twelve rivers and measured a similar value for $R$ of 1.8 (95% CI, 1.41 to 2.20, Fig. 3c) at an even lower value for $F_N$ (see Table 1). Note, we might have expected $R$ to increase steeply as an inverse function of $F_N$ (Supplementary Figure 3). We can predict what values for $R$ we might have expected if our N$_2$ gas had been produced by either denitrification or anammox fuelled by porewater nitrite and/or ammonia, respectively (Fig. 2a) and compare them to our

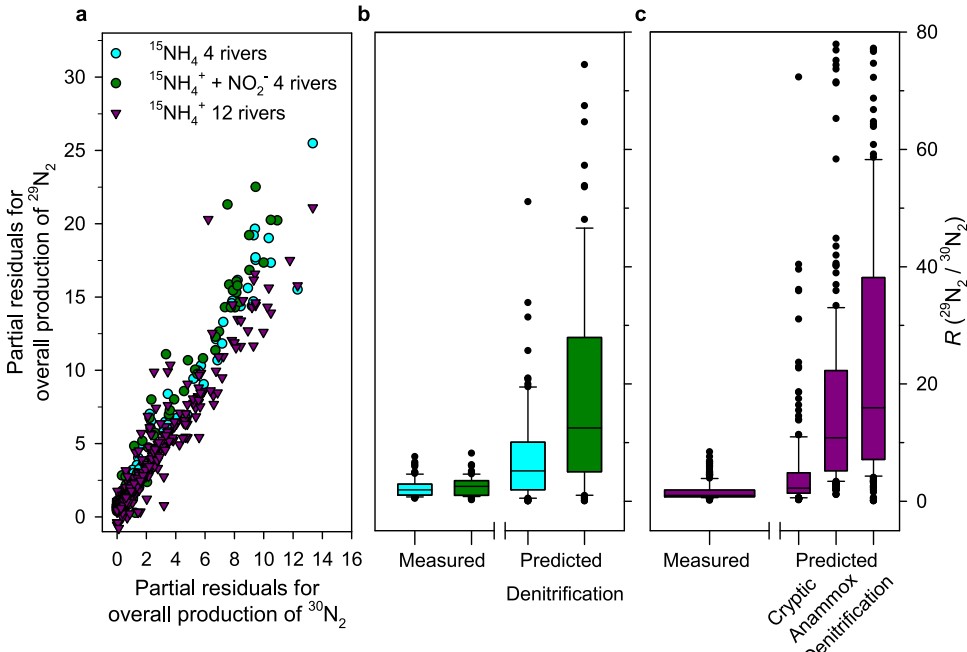

**Fig. 3 Ratios of $^{29}N_2$ and $^{30}N_2$ production consistently below those predicted. a** Consistent $^{29}N_2$ production (nmol g$^{-1}$ h$^{-1}$) from $^{15}$N-ammonia added to oxic sediments, against each corresponding measure of $^{30}N_2$ production at each time-point (>0.5 h < 10 h) in each incubation in Fig. 1a presented here as the partial residuals from mixed-effects models ($n = 100$ and $n = 300$, for the 4- and 12-river datasets, respectively). **b** The corresponding measured values for $R$ from **a**, for the first 4 rivers incubated with either $^{15}NH_4^+$ (95% CI for $R = 2.01$ to 2.64) or $^{15}NH_4^+$ and additional $^{14}NO_2^-$ (95% CI for $R = 2.12$ to 2.74), against those predicted for denitrification of porewater $NO_2^-$. **c** Measured $R$ values for the 12 river sediments incubated with only $^{15}NH_4^+$ (95% CI for $R = 1.41$ to 2.20), against predicted $R$ values for denitrification, anammox, and a cryptic coupling. See main text and Table 2. Upper and lower box boundaries are 75th and 25th percentiles, respectively, upper and lower whiskers are 90th and 10th percentiles, respectively, the extreme outliers the maxima and minima and the horizontal line the centre, median value.

**Table 2 Summary of the overall measured and predicted ratios of $^{29}N_2$ to $^{30}N_2$ production ($R$) for treatments 2 and 4 and the fraction of $^{15}$N labelling in each substrate pool for $F_N$ and $F_A$, on average. Overall measured and predicted $R$ estimates, standard errors (s.e.) and 95% confidence intervals were derived with mixed-effects models using lme4 and emmeans (See Methods) and similarly for $F_N$ and $F_A$. See Supplementary Table 3 for further details.**

| Code, Treatment | Rivers (replicates) | $R$ ($^{29}N_2/^{30}N_2$) | Lower 95% C.I. | Upper 95% C.I. | $P$ | $F_N^{\ddagger}$ | | $F_A^{\ddagger}$ |
|---|---|---|---|---|---|---|---|---|
| | | *Measured* | | | | $NO_2^-$ | $NO_x^-$ | |
| **2**, $^{15}NH_4^+$ | 4 (5) | 2.32 (0.16) | 2.0 | 2.6 | | 0.32 | 0.41 | 0.57 |
| **4**, $^{15}NH_4^+ + ^{14}NO_2^-$ | 4 (5) | 2.43 (0.16) | 2.1 | 2.7 | | 0.27 | 0.36 | 0.51 |
| **4 minus 2** | 4 (5) | 0.11 (0.08) | | | 0.20 | | | |
| **2 & 4** | 4 (10) | 2.38 (0.15) | 2.1 | 2.7 | | | | |
| **2**, $^{15}NH_4^+$ | 12 (5) | 1.81 (0.20) | 1.4 | 2.2 | | 0.16 | 0.25 | 0.45 |
| | | *Predicted$^{\dagger}$* | | | | | | |
| **2**, $^{15}NH_4^+$ | 4 (5) | 7.81 (1.36)$D$ | 5.1 | 10.5 | | | | |
| **4**, $^{15}NH_4^+ + ^{14}NO_2^-$ | 4 (5) | 20.05 (1.49)$D$ | 16.9 | 22.3 | | | | |
| **2**, $^{15}NH_4^+$ | 12 (5) | 29.4 (2.28)$D$ | 24.8 | 33.9 | | | | |
| **2**, $^{15}NH_4^+$ | 12 (5) | 19.3 (2.28)$A$ | 14.8 | 23.9 | | | | |
| **2**, $^{15}NH_4^+$ | 12 (5) | 9.3 (2.28)$C$ | 4.8 | 13.8 | | | | |

$^{\dagger}$Predicted $R$ values, using Eqs. (6) and (7), for denitrification ($D$), anammox ($A$) and a cryptic ($C$) processes fuelled by porewater $NO_2^-$, see Supplementary Table 3 for $NO_x^-$. Note also that the predicted values are derived using each individual measure of $F_N$ and $F_A$ in each vial, and that $F_N^{\ddagger}$ are the overall mean values simply to illustrate the effect of adding $^{14}NO_2^-$ to the incubations with sediments from the first 4 rivers and overall lower $F_N$ value for the 12 river incubations.

measured $R$ values to highlight the disparity between the two (Fig. 3b, c and Table 2):

$$\text{Predicted } R \text{ for denitrification,} \quad R = \frac{2 \times F_N \times (1 - F_N)}{F_N^2} \quad (6)$$

$$\text{Predicted } R \text{ for anammox,} \quad R = \left(\frac{1}{F_N} - 1\right) + \left(\frac{1}{F_A} - 1\right) \quad (7)$$

Our measured $R$ values were too low to be explained by either denitrification or anammox fuelled by porewater $F_N$ and/or $F_A$ (Fig. 2a) and even a mixture of these two processes couldn't produce such low values for $R$ on average. This consistent disparity between our measured and predicted values for $R$, according to the accepted model, along with the constancy in $R$, despite differences in $F_N$ (Table 2), strongly implies that porewater $NO_x^-$ had little influence on the $^{15}$N-labelling of the

$N_2$ gas produced from the oxidation of $^{15}$N-ammonia. Further, in an analogous set of incubations where we added $^{15}$N-nitrite instead of $^{15}$N-ammonia, we measured no consistent production of $^{15}$N-$N_2$ gas (Treatments 5 & 6 Table 1 and Methods). Hence, nitrogen for $N_2$ formation was not drawn primarily from the porewater $NO_x^-$ pool (Fig. 2a). Instead, we propose that any $N_2$ producing pathways draw from a cryptic nitrogen pool (Fig. 2b) with $^{15}$N-labelled fraction, $F_{Ncry}$, instead of the familiar porewater pool with $^{15}$N-labelled fraction, $F_{Npw}$. Indeed, if we invoke a cryptic pool by making the $^{15}$N-labelling of $F_N$ the same as $^{15}$N-ammonia in the porewater $F_A$ in Eqs. (6) and (7) and thereby force denitrification and/or anammox to draw on that $F_{Ncry}$ pool, then the predicted $R$ values come closer to our measured $R$ values ($R$ cryptic, Fig. 3c and Table 2).

**$N_2$ is produced from ammonia through a cryptic intermediate.** We can use both the accepted[21] and a new mathematical framework to more formally justify our proposal for a cryptic intermediate pool or process. First, we define the proportion of $N_2$ gas coming from anammox relative to denitrification that is conventionally known as $ra$[15]. $ra$ has to lie between 0 and 1 and, in the accepted framework, is expressed as a function of porewater $F_A$ and $F_N$ and $R$ according to[21] (See Eq. (1) to (14) in Supplementary Note 1):

$$ra = \frac{(R+2) \times F_N^2 - 2 \times F_N}{(F_N - F_A) \times [(R+2) \times F_N - 1]} \quad (8)$$

In the accepted framework, however, our measured values for $R$ and porewater $F_A$ and $F_N$ generate nonsensical estimates for $ra$ (e.g., $-6.06$ to $3.03$, not $> 0 < 1$). Just as for Fig. 3c, we cannot apportion $N_2$ gas between anammox and denitrification drawing on porewater $F_N$ and/or $F_A$ – in the conventional sense – to produce our measured $R$ values (Fig. 2a). Next, we define the $^{15}$N-labelling of the $N_2$ gas produced ($F_{N2}$), which, like $ra$ (Eq. 6), also has to lie between 0 and 1 (See Eq. (1) to (14) in Supplementary Note 1).

$$F_{N2} = F_N - \frac{R \times F_N + 2 \times (F_N - 1)}{2 \times (R + 2 - \frac{1}{F_N})} \quad (9)$$

Unlike $ra$, which is expressed as a function of both porewater $F_A$ and $F_N$, only $F_N$ is required to parameterise $F_{N2}$ (Eq. 9 $cf.$ Eq. 8). That is not to say that $F_A$ has no influence on $F_{N2}$, as $F_N$—be it either the $F_{Ncry}$ or $F_{Npw}$ pools—must result from ammonia oxidation drawing on $F_A$ (Fig. 2).

We can then use solutions to Eqs. (8) and (9) between $> 0 < 1$ to define a solution space for any combination of $F_N$, $F_A$, and realistic values for $R$ (See Supplementary Figure 3 for $R$ as a function of $^{15}$N atom %) that we can visualise as a 3D ribbon (Fig. 4). The height of the ribbon is defined in terms of $F_{N2}$ and is depicted here for our average value for $F_A$ of 0.51 (Table 1 and see Supplementary Fig. 4 for $F_A$ at 0.1 and 0.9). Overall the ribbon is very narrow and where $F_A = F_N$ there are no solutions and this singularity appears as a gap in the ribbon. If $F_{Ncry}$ is isolated and derives solely from the oxidation of $F_A$ (Fig. 2b), then $F_{Ncry}$ has to equal $F_A$. Further, if $F_{N2}$ is only dependent on $F_N$ (Eq. 9) and this $F_N$ is equivalent to $F_{Ncry}$, then our calculated values for $F_{N2}$—plotted as functions of our measured values for $R$ and $F_A$ (where $F_{Ncry}$ equal $F_A$)—should fall near the gap in the ribbon where $F_N$ equals $F_A$. This is indeed what we observe and especially for the better parameterised 12 river estimate (Fig. 4). In contrast, if we again force denitrification to be the only source of $N_2$, and calculate $F_{N2}$ assuming that $F_N = F_{Npw}$ (Fig. 2a), then the points fall away from our measured $R$ values. Hence, in the presence of $^{15}$N-ammonia and oxygen, our measured $R$ values only make sense if we assume $F_{Ncry} = F_A$ (Fig. 2b) i.e., the porewater nitrite

pool essentially represents the left-overs of the cryptic transformations during which $N_2$ is produced.

**Internal $NO_x^-$ cycling or a novel pathway or organism.** We propose that the coupling between ammonia oxidation and $N_2$ gas production in oxic, permeable riverbed sediments involves a cryptic intermediate pool derived solely from the oxidation of ammonia that remains isolated from the porewater prior to the production of $N_2$ gas. In one scenario, a cryptic pool, similar to the porewater $NO_x^-$ pool, is fed by the oxidation of ammonia to $NO_x^-$, or possibly NO (ref. [3,23,24]), through nitrification. The pathway from $F_{Ncry}$ to the production of $N_2$ gas, however, branches off before that $NO_x^-$ mixes with the ambient porewater $NO_x^-$ (Fig. 2b) and would require internal $NO_x^-$ cycling. Internal $NO_x^-$ cycling is recognised as a potential source of interference for $^{15}$N isotope tracer studies in the ocean[25,26] and is known in the consortia of ammonia oxidisers and anammox bacteria in wastewater CANON[27] reactors (Complete Autotrophic Nitrogen removal Over Nitrite. Figure 2b, reactions 1 & 4) – though the actual mechanism in nature remains unknown.

Alternatively, some aerobic ammonia oxidising bacteria first produce nitrite (reaction 1) that they then reduce to $N_2O$ gas in a process known as nitrifier-denitrification[3]. Known nitrifier-denitrifier bacteria, however, lack a canonical $N_2O$-reductase (NOS, $nosZ$) to reduce $N_2O$ to $N_2$ gas, so are not currently recognised as complete denitrifiers (reaction 7, Fig. 2b). Nitrosocyanin, a soluble red Cu protein isolated from *Nitrosomonas europaea*[28], is recognised as a plausible substitute to canonical $N_2O$-reductase that could enable complete nitrifier-denitrification to $N_2$ gas[3]. Our data enable us to test this hypothesis. For example, we know that $^{15}NO_2^-$ from the initial oxidation of $^{15}NH_4^+$ exchanges with the porewater (reaction 1, Figs. 1b and 2a) and we would expect, therefore, that $^{15}NO_2^-$ added to the porewater would be available to any nitrifying-denitrifying bacteria[29]. We have, however, already shown that adding $^{15}NO_2^-$ to the porewater resulted in no consistent production of $N_2$ gas (Treatments 5 & 6, Table 1) i.e., $N_2$ gas production is dependent on the initial oxidation of $^{15}$N-ammonia. This fact, along with the clear discrepancy between the measured and predicted scenarios involving porewater $NO_x^-$ (Figs. 3b, 3c & 4) make it hard to reconcile our $N_2$ gas production with either nitrifier-denitrification or canonical denitrification (reactions 3a, 3b & 7, Fig. 2).

Finally, it is theoretically possible for ammonia to be completely oxidised by oxygen to $N_2$ gas (equation 5[8]) within a single, unknown organism. Such a reaction offers the simplest explanation for our results, with their strong dependency on aerobic ammonia oxidation and lack of influence from external porewater nitrite. Regardless of the actual pathway that produces the $N_2$ gas (Fig. 2b), an isolated cryptic intermediate pool has to have the same $^{15}$N-labelling of the ammonia pool ($F_{Ncry} = F_A$). As a consequence of this equality, we can no longer distinguish between sources of $N_2$ gas, be it a denitrification-like pathway reductively combining N from an oxidised cryptic pool, an anammox-like process drawing on ammonia and cryptic N, or complete ammonia oxidation, as they would all produce $^{29}N_2$ and $^{30}N_2$ at the same ratio (Fig. 2b where $R$ is equal for each process).

Our observations challenge the current understanding of a key coupling in the nitrogen cycle in permeable, oxic riverbed sediments that may also apply to other biomes where the oxidation of ammonia is tightly coupled to the production of $N_2$ gas, such as continental shelf-sediments[30,31] and groundwater aquifers[17]. Whether it transpires that our cryptic coupling is mediated by a novel organism or, as of yet, a masked combination of known players in the nitrogen cycle remains to be resolved.

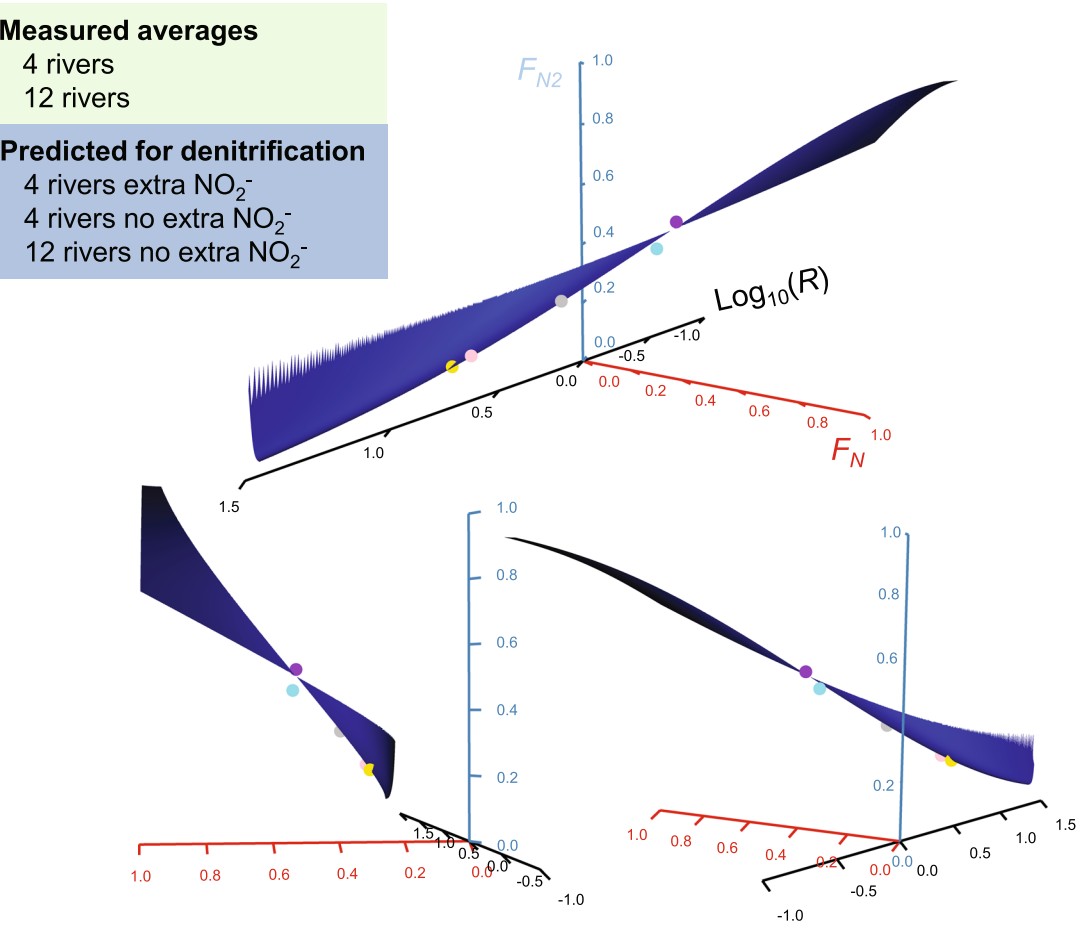

**Fig. 4 Orientations of the solution space ribbon with both measured and predicted values for $R$.** Here we present all data in just one solution space for the average fraction of $^{15}N$ in the ammonia pool ($F_A$) of 0.51 and combinations of Eq. (8) ($F_{N2}$) and 9 ($ra$) both yielding values between $> 0 < 1$. $R$ is the ratio of $^{29}N_2$ to $^{30}N_2$ and $F_N$ and $F_{N2}$ the fraction of $^{15}N$ in the $NO_x^-$ and $N_2$ gas pools, respectively. To plot $F_{N2}$ for each of our measured values of $R$ we have to assume that $F_N$ equals $F_A$ measured in the porewater. In the solution space, there are no solutions where $F_A = F_N$ (i.e., 0.51) and this singularity appears as a gap in the ribbon. Despite measurable changes in porewater $F_N$, the average values for both the 4-river and 12-river study appear near to each other and the gap where $F_A = F_N$. Note that the better parameterised 12-river average touches the gap and by inference, $F_A \approx F_{Ncry}$ (Fig. 2b). Denitrification fuelled by porewater $NO_x^-$ predicts values away from our measured values for $R$. Note, for the single predicted denitrification $R$ values we use the median $F_N$ values.

## Methods

**Study sites and sediment sampling**. We began by collecting sediment samples from four rivers which we subsequently widened to a total of twelve rivers in southern England, UK, between October 2015 and May 2016 (Supplementary Figure 1 and Supplementary Table 1). Among them, the Rivers Lambourn, Darent, Wylye, Rib, Pant, Stour (1) and Stour (2) have chalk-based, permeable gravel-dominated riverbeds, while the Rivers Marden, Hammer, Medway, Broadstone, and Nadder have less permeable, sand-dominated riverbeds as described elsewhere[18,32,33]. At each river, surface sediments (<5 cm) were collected from five different locations using Perspex corers (13-cm × 9-cm internal diameter, 827 mL and sealed at one end with an oil-seal stopper)) which were transferred to plastic zip-lock bags (VWR International) and stored in a cool bag (Thermo) during transport back to the laboratory. Each sediment sample from each river was then homogenised in the laboratory for the experiments described below.

*Aerobic ammonia oxidation in oxic sediment slurries.* $^{15}N$-$NH_4^+$ oxidation experiments were carried out with sediments first from four rivers (the rivers Lambourn, Wylye, Marden, and Hammer) and then all twelve. In a standard anoxic application of $^{15}N$ isotope pairing techniques[34–36], ambient porewater nitrite, nitrate, and any residual oxygen are removed by pre-incubating the anoxic sediment slurries for 12 h to 24 h before adding any $^{15}N$-tracers[35,36]. Here this was not possible as we were measuring the aerobic oxidation of $NH_4^+$ and so to avoid contamination from the high background $^{14}NO_x^-$ ($^{14}NO_3^-$ + $^{14}NO_2^-$), which is typical for these rivers[24], instead we used nitrite- and nitrate-free synthetic river water (0.12 g/l $NaHCO_3$, 0.04 g/l $KHCO_3$, 0.07 g/l $MgSO_4 \cdot 7H_2O$, 0.09 g/l $CaCl_2$ $2H_2O$, pH = 7) to make the sediment slurries as before[18].

Oxic slurries were prepared by adding approximately 3 g sediment (~0.75 ml of porewater) and 2.7 ml air-saturated synthetic river water into 12 ml gas-tight vials (Exetainer, Labco), leaving an approximate 6 ml headspace of air which is equivalent

to ~58 µmol $O_2$ per prepared vial. We know from previous incubations with similar sediments from 28 rivers[37] respiration rates to be ~187 nmol $O_2$ g$^{-1}$ h$^{-1}$, on average (±64.3, 95%, C.I.), that would consume ~12% of the total oxygen during a 12 h incubation. In addition, we also checked oxygen over time using a microelectrode (50 µm, Unisense) in parallel sets of scaled-up slurries (120 mL with the same ratio of sediment to water to headspace) for two rivers and found comparatively little consumption as before[18] and see example in Supplementary Figure 2.

To trace the oxidation of ammonia to $N_2$ gas, the prepared oxic slurry vials were then sealed and injected with 100 µl of 14 mM $^{15}NH_4^+$ stock-solutions (98 atom% $^{15}N$, Sigma-Aldrich) to generate final porewater concentrations of ~390 µM $^{15}NH_4^+$. This high $^{15}N$ concentration ensured sufficient labelling of the ammonia pool (~50%) to enable quantifiable production of both single-labelled, $^{29}N_2$, and dual-labelled, $^{30}N_2$, in order to calculate $R$ in Eqs. (6) to (9). To link the production of $N_2$ gas to the initial aerobic oxidation of ammonia, an additional set of slurries were injected with 100 µl of 14 mM $^{15}NH_4^+$ (as above), along with 2.8 mM (stock-solution) of the ammonia mono-oxygenase inhibitor[19], allylthiourea (ATU), to give final porewater concentrations of ~390 µM $^{15}NH_4^+$ and ~80 µM ATU. While we have shown previously that 80 µM ATU inhibits aerobic ammonia oxidation in gravel and sandy riverbed sediments[18], higher concentrations maybe required in other settings[38]. All of the oxic slurry vials were then incubated on a shaker (120 rpm, Stuart SSL1) for up to 12 h (Table 1, Treatments 1 and 2) in a temperature-controlled room at 12 °C. Incubations amended with just $^{15}NH_4^+$ were terminated at 0 h, 0.5 h, 1 h, 3 h, 4.5 h, 6 h, 9 h, and 12 h while those amended with both $^{15}NH_4^+$ and ATU were terminated at 0 h, 3 h, 6 h, and 12 h by injecting 100 µl of formaldehyde (38%, w/v) through the vial septa. All vials were then stored upside down prior to quantification of $^{29}N_2$ and $^{30}N_2$ by mass-spectrometry and $R$ is then simply $^{29}N_2/^{30}N_2$ (see below).

In addition to measuring the production of $^{29}N_2$ and $^{30}N_2$ gases ($R$), the fraction of $^{15}N$ in the inorganic nitrogen porewater pools ($F_A$ for ammonia and $F_N$

for $NO_x^-$ e.g., $NO_2^-$ plus $NO_3^-$) needed to be quantified too (see Eqs. 6 to 9). To avoid any potential interference from formaldehyde, on the analysis of the inorganic nitrogen species, a parallel set of $^{15}NH_4^+$ amended slurries was prepared solely for nutrient analyses. At each time point (as above for $N_2$ gas analysis), vials were injected with 20 µL of 1.6 M NaOH to preserve nitrite before being frozen at −20 °C[39]. Samples were defrosted and centrifuged at 1200 rpm for 10 min and the collected supernatant analysed (see below).

*Manipulating the degree of $^{15}N$-labelling in the porewater $NO_2^-$ pool* ($F_N$ as $F_{Npw}$). In typical anoxic sediment slurry incubations used to quantify $N_2$ gas production from denitrification and anammox[34,35], the fraction of porewater substrate labelled with $^{15}N$ ($F_A$ or $F_N$) influences the ratio of $^{29}N_2$ to $^{30}N_2$ produced. To characterise the influence of porewater $NO_2^-$ on the coupling between $^{15}N$-$NH_4^+$ oxidation and $^{15}N$-$N_2$ production in oxic sediment slurries, we manipulated the fraction of porewater $NO_2^-$ labelled with $^{15}N$. Oxic sediment slurries from the first four riverbeds were injected (100 µl) with combinations of stock-solutions of 14 mM $^{15}NH_4^+$ and 840 µM $^{14}NO_2^-$ or just 14 mM $^{15}NH_4^+$ and both with or without 2.8 mM ATU. This generated final porewater concentrations of ~390 µM $^{15}NH_4^+$, ~24 µM $^{14}NO_2^-$ or ~80 µM ATU and the prepared vials were then incubated on a shaker as above (see Table 1, Treatments 3 and 4). As above, oxic slurry vials were sacrificed at different time points for $^{15}N_2$ gas analysis and with a parallel set of $^{15}NH_4^+$ or $^{15}NH_4^+$ plus $NO_2^-$ amended slurries solely for nutrient analyses.

To further test the dependency of $N_2$ gas production on the initial oxidation of $^{15}N$-ammonia, we also performed a set of analogous incubations with sediments from the first four rivers with $^{15}NO_2^-$ (Table 1, Treatments 5 and 6). Here everything was the same (amount of sediment, with or without ATU, incubation times, oxygen etc.,) except the $^{15}N$-labelling was added with nitrite rather than ammonia (as above) to final concentrations of ~390 µM $^{14}NH_4^+$ and ~24 µM $^{15}NO_2^-$ (98 atom% $^{15}N$, Sigma-Aldrich). If active, we would have expected $N_2$ gas production from reactions 3b and 4.

*Analytical methods*. Headspaces of the oxic slurry samples were analysed for $^{15}N$-$N_2$ using a continuous-flow isotope ratio mass spectrometer (Sercon 20–22, UK) as described elsewhere[18]. The mass spectrometer has a sensitivity of 0.1 ‰ $^{15}N$ which here translates to approximately 0.1 nmol $^{15}N$-$N_2$ $g^{-1}$ dry sediment. To determine porewater $F_N$ ($NO_2^-$ or $NO_x^-$, below) the concentration of $^{15}NO_2^-$ in the $^{15}NH_4^+$ treatments was measured, the preserved supernatants were diluted and 3 ml of sample transferred into a new 3 ml gas-tight vial (Exetainer, Labco), the vial capped and a 0.5 ml helium headspace (BOC) added. Samples were injected with 100 µl of sulfamic acid (4 mM in 4 M HCl) and placed on a shaker (120 rpm, Stuart SSL1) overnight to reduce $^{15}NO_2^-$ to $^{15}N$-$N_2$ and the headspaces subsequently analysed for $^{15}N$-$N_2$ as above[18,40]. For $^{15}NO_x^-$ ($^{15}NO_2^-$ plus $^{15}NO_3^-$) analysis, 0.3 g spongy cadmium and 200 µl of 1 M imidazole, along with 3.5 ml of sample were added to each gas-tight vial (12 ml, Exetainer, Labco) and the vials shaken (120 rpm, Stuart SSL1) for 2.5 h to reduce $^{15}NO_3^-$ to $^{15}NO_2^-$ and the samples then treated as above to convert $^{15}NO_2^-$ to $N_2$[18,41]. The sensitivity for $^{15}NO_x^-$ was approximately 0.4 nmol $^{15}N$ $g^{-1}$ dry sediment. $F_N$ was then calculated for $NO_2^-$ or $NO_x^-$ as:

$$F_N = \frac{^{15}NO_x^-}{\left(^{15}NO_x^- + \,^{14}NO_x^-\right)} \qquad (10)$$

And similarly for $F_A$:

$$F_A = \frac{^{15}NH_4^+}{\left(^{15}NH_4^+ + \,^{14}NH_4^+\right)} \qquad (11)$$

Where $^{15}NH_4^+$ was determined by the increase in concentration, measured by standard indophenol-blue wet-chemistry, above ambient background in controls after the addition of $^{15}NH_4^+$.

Sediment particle size was determined by sorting the dried sediments through a series of sieves (Endecotts Ltd, England) from 16 mm, 13.2, 8, 4, 1.4, 0.5, 0.25, 0.125, to 0.0625 mm and then weighing each size fraction. Grain size distributions were calculated and classified on the Wentworth scale as gravel (particles coarser than 2 mm), sand (particles between 0.0625 and 2 mm), mud (silt plus clay material finer than 0.0625 mm)[42]. For sediment organic C and N content, disaggregated samples were oven-dried, acidified by HCl (2 M) to remove inorganic carbonates[43] and re-dried to a constant weight. Then ~50 mg of sediments were transferred to tin-cups, reweighed, and combusted at 1000 °C in an integrated elemental analyser and mass-spectrometer (Sercon, Integra 2, UK).

*Statistical analysis*. We used mixed-effects models to estimate overall rates of total $^{15}N$-$N_2$ gas production during the incubations (Fig. 1a), treating each of either the first four or subsequent twelve rivers as genuine, independent replicates. Within each river, each of the 5 technical replicates were nested within each respective river and fitted as random effects on the slope and intercept in each case; though it was not always necessary to retain replicate or all the random effects in a model to get the best fit to the data – based on lowest AIC (Akaike Information Criterion). To visualise the consistent production of $^{29}N_2$ to $^{30}N_2$ across the incubations with $^{15}N$-ammonia, we regressed each measure of $^{29}N_2$ on each measure of $^{30}N_2$, at each time point, in each incubation and display (Fig. 3a) the partial residuals for the best fitting model[44]. To estimate the overall average measured and predicted ratios of

$^{29}N_2$ to $^{30}N_2$ (R) we only used the data for the time points >0.5 h < 10 h i.e., when there was measurable (~0.1 nmol $N_2$ $g^{-1}$ dry sediment), steady-production of both $^{15}N$ labelled gases, divided each measure of $^{29}N_2$ by each respective measure of $^{30}N_2$ at each time point, in each incubation and treated river and replicate as above. For the first 4 rivers, the ratio R was estimated by fitting each time point as a random-effect, but for the larger, 12 river dataset, time was fitted as a fixed-effect and R estimated for the middle time point in the incubations and similarly for $F_N$ (for both $NO_2^-$ and $NO_x^-$) and $F_A$. All statistical analyses were performed in R (version 3.6.3, 2020-02-29) under RStudio (version 1.2.5033). Model fitting was carried out in the "lme4" package (version 1.1-21) and parameter (marginal mean) estimates, standard errors, and confidence intervals derived using the "emmeans" package (version 1.4.5) with Kenwood-Roger degrees of freedom and Tukey correction where appropriate.

## Data availability
Source data are provided with this paper.

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

## Acknowledgements

We thank Ian Sanders and Katrina Lansdown for technical assistance and Axel Rossberg for help with the 3D imagery and Queen Mary University of London and the Chinese Scholarship Council for supporting the research.

## Author contributions

M.T. and L.O. conceived the study and L.O. performed all of the experiments and B.T. formulated the mathematical framework. L.O. and M.T. analysed the data and M.T. and B.T. drafted the manuscript. All authors commented on and revised the manuscript.

## Competing interests

The authors declare no competing interests.
