## [Peer Review File · Nature Communications]

Reviewers' comments:

Reviewer #1 (Remarks to the Author):

In "Coupled nitrification and N₂ gas production as a cryptic process in oxic riverbeds," Ouyang et al present nice experimental data indicating that when 15N-NH₄⁺ spikes were used to calculate rates of N₂ production, the intermediate, nitrite, did not mix in the sediment pore waters. Thus the basic assumption of the technique that is widely used to obtain N₂ production rates was found to be untrue. While this paper does an excellent experimental job of proving the presence of this problem, the problem, or a very similar problem with 15N-NO₂⁻, has already been noted and explained in previous papers (see below). I believe that the fact that sediments are composed of particles can explain the author's data and this should be addressed in the paper. Proposing the activity of a novel player in the N cycle is unnecessary. Overall, after editing, this paper will be a good addition to the literature that will highlight a fundamental problem with a widely used technique.

Issue 1: ATU (allylthiourea) may be an inhibitor of long standing, but it was originally tested before we knew about ammonia oxidizing archaea. The authors need to convince the reader that ATU inhibits the organisms that they say it does and not others. I shouldn't have to figure it out for myself.

1) Martens-Habbenha et al 2015 Environmental Microbiology 17: 2261-74.

Martens_Habbenha et al 2015 indicate that ammonium oxidizing archaea were 97% inhibited at 330 μM ATU. Betaproteobacterial ammonia oxidizing bacteria were inhibited at 10 μM ATU and gammaproteobacterial ammonia oxidizing bacteria were inhibited at 100 μM ATU.

The present paper used 2 mM ATU, so everything is inhibited 10x over.

However, does ATU inhibit anammox bacteria??

2) Dapena-Mora et al 2007 Enzyme and Microbial Technology 40: 859-865.

Not an amazing paper, but it indicates that 1 g/L ATU does not affect anammox bacteria.

3) van de Graaf et al 1996 Microbiology 142: 2187 also says that ATU does not inhibit anammox.

Issue 2: Similar phenomena in oceanic oxygen deficient zones using 15N-NO₂⁻ have published previously in three papers. These papers should all be cited.

Both De Brabandere et al 2014 Environmental Microbiology 16: 3041-3054.

and

Chang et al 2014 Limnology and Oceanography 59: 1267-74 find an excess of 29N₂ produced in enriched 15N experiments with nitrite compared to experiments with 15N-NH₄⁺.

Basically, as the authors noted in the present paper, in these two oxygen minimum zone papers, the isotopic composition of labeled N₂ cannot be explained by the binominal distribution when anammox is taken into account. In Chang and De Brabandere, there is too much 29N₂ for the amount of denitrification that should be present. Both papers explain this data by suggesting an internal nitrite pool for denitrifiers. Which is to say, that the nitrite used by denitrifiers is not mixing with the spiked surrounding water. Exactly as seen in the current paper in pore waters.

Nicholls et al 2007 also see a similar phenomena. (co-authored by authors of the current paper but not cited in the current paper)

Internal nitrite pools have been indicated in denitrifiers in a sulfidic fjord by Jensen et al 2009 Marine Chemistry 113: 102-113.

Fuchsman et al 2018 Deep Sea Research II 156: 137-147 examined natural stable isotopes of N₂ and nitrite and nitrate. The fractionation factor from (nitrate and nitrite) to N₂ gas changed with depth in the oxygen minimum zone. Only two hypotheses could explain this phenomenon: an internal nitrite pool in denitrifying bacteria or an internal nitrite pool inside of sinking particles. Mathematically the results of these two situations are similar.

Particles have diffusive boundary layers that separate their insides and outsides (Ploug et al 1997 Aquatic Microbial Ecology 13: 285-294) and often have elevated nutrients inside the particles (Simon et al 2002 Aquatic Microbial Ecology 28: 174-211).

Wilson et al 2014 Deep Sea Research I 114: 47-55 has shown elevated nitrite in oxic particles in the ocean.

In ocean oxygen minimum zones, denitrifiers are enriched on particles (Ganesh et al 2015 ISME 9: 2682-2696 and Fuchsman et al 2017 Frontiers in Microbiology 8: 2384.). It is likely that internal nitrite pools in particles can explain this entire phenomena including data from Chang et al 2014 and De Brabandere et al 2014.

Issue 3. The data in the current paper can be explained by particles.

The current paper takes place in sediments. Sediments are composed of particles. We already know that the fact that sediments are composed of particles greatly affects N cycling there. Particles are the explanation why the natural fractionation of N₂ production in sediments is close to zero. Nitrate is completely consumed in the particles. Implicit in this accepted argument is the idea that the inside of particles and the pore waters are not exchanging quickly. See:

Brandes and Devol 1997 Geochimica et Cosmochimica Acta 61: 1793-1801

Lehmann et al 2007 Geochimica et Cosmochimica Acta 71: 5384-5404.

For this to explain the current data, I believe that spiked ammonium would have to diffuse into particles but nitrite would not diffuse back out. It becomes part of an internal nitrite pool. This pool could be inside the particle or inside an organism. The experiments in the current paper were 8 hours long—pretty short. Probably a longer incubation would have more exchange.

Ammonium is a precious item that organisms want.

Particles also explain anaerobic processes under oxygen. This entire phenomenon is related to particles and particles need to be addressed in the paper.

See Steif et al 2016 Frontiers in Microbiology 7: 98.

Ploug and Bergkvist 2015 Marine Chemistry 176: 142-149.

Bianchi et al 2018 Nature Geoscience 11: 263.

Issue 4. Given that the data in the paper can be explained by denitrification in sediment particles, which we already know happens, it seems like invoking a new player (or coupling) in the N cycle is unnecessary.

Detailed comments:

Line 51: "Similarly, aerobic nitrification can fuel both anaerobic denitrification⁹ and anammox¹⁰ at the margins of the vast oceanic oxygen minimum zones"

Here ref 9 is Ward et al 2009—that paper has absolutely nothing to do with aerobic nitrification fueling denitrification. I just reread to be sure.

Reference 10 is Lam et al 2007, which is about aerobic nitrification fueling anammox (specifically) in the Black Sea. The Black Sea is fundamentally different from oceanic oxygen minimum zones in that it has a large flux of ammonium coming from its underlying sulfide zone. If you would like to see these fluxes quantified see Fuchsman et al 2008. On the other hand, in the giant oceanic oxygen minimum zone, ammonium is below detection (<10 nM) in the most of the oxygen minimum zone. See both Widner et al 2018s for ammonium data in the ETNP and ETSP.

Bristow et al 2016 indicates that the K_m for oxygen for aerobic ammonium oxidation is higher than the K_m for nitrite oxidation. Rather than aerobic ammonium oxidation fueling denitrification/anammox in the oceanic oxygen minimum zones, nitrate reduction coupled to nitrite oxidation is a key process using the small concentrations of oxygen. Also note that, unlike the Black Sea, in OMZs nitrate concentrations are >20 μM from ambient seawater. aka oxidized N is not a limiting factor for denitrification.

Please delete.

Also this means that the authors have to reword that entire paragraph.

Line 96: This paper uses a lot of math. It might help the reader to follow if instead FA from fraction of ammonium, you used F_{NH3} and F_{NOx} because FA could also be fraction anammox. It is a simple change that could make things easier to follow.

Perhaps Table 1 should include the oxygen concentrations for each experiment.

Line 188: Here we go again. The Black Sea is not like the margins of oxygen minimum zones due to the large flux of ammonium into the Black Sea suboxic zone. Both references 10 and 30 are about the Black Sea. The Kalvelage et al 2011 reference is more legitimate. However, Kalvelage assumes rather than shows that aerobic ammonium oxidation and anammox are linked. If you want to say this, you should really be citing Lam et al 2009 PNAS. That is where this idea is originally coming from. In that paper it comes from a calculation based on the balance of rates. However, back in the day of the Lam paper, the rubber stoppers were infecting denitrification rate measurements with oxygen and inhibiting them, so that calculation is very probably incorrect.

If you look at models by Zakem et al 2020 ISME 14: 288-301—like Fig 3—you will see that aerobic ammonium oxidation to nitrite is not a key player in the OMZ system or its edge.

You don't need to cite oxygen minimum zones for this process to be important. Sediments are quite important by themselves.

Reviewer #2 (Remarks to the Author):

This paper uses isotope pairing methods on oxic riverine sediments in a lab to deduce that there is a cryptic intermediate pool of oxidized N that is not part of the measurable porewater pool. If true, this is a tremendously interesting finding. My criticisms of the paper have nearly nothing to do with the "if true" part. This is high-risk, high-reward science and as such I support it fully. Methods and modeling are as good as they can be to address this question in this paper. My criticisms are with manuscript presentation and explanation of how the authors link that data and mathematical framework to new understanding. Thus my comments below are meant to improve the manuscript and not to argue against its publication.

I struggled to follow along with the paper from the methods the data and the mathematics. There are two reasons for this. 1. This work is inherently highly technical and never is going to be easy to understand. Despite 25 y of N isotope experience and a recent foray into isotope pairing in my lab, this paper took a lot of work to review. The authors may wish that I were smarter, and I would agree with them. But I am not, so 2. I think that there is a lot of room for clarifying the paper to enable readers to follow along with the reasoning. This clarification can include: mathematical notation, explanation of methods, figures. The storyline of the paper is very much falls along how the authors discovered it: data followed by modeling to see what is possible and then a conclusion. Would putting the modeling first followed by data that supports (or not) the models work better? I have a lot of comments below.

Consider some sort of diagram that outlines the methods and how they lead to a conclusion. A flowchart may work, but this is not to be prescriptive. But I think at any way to lead the reader from the methods to results to understanding would be a big help.

This paper lives and dies on its methods. The authors have done new techniques to putatively show new N pathways. But the methods section is written very tersely for few specialists who do these methods. I think that there should be more text linking what the authors want to know with how the method addresses this point. I admit to having trouble making these points myself despite having just starting using isotope pairing methods in my lab to attempt to measure denitrification in oxic aquifers. The Results section has a lot of this text and yes, it covers some of the methods there, great. But I would spell out these details in both places.

The arguments for what is actually happening to drive these patterns (starting on line 171) are critical parts of the paper. But I note with interest how the discovery of these pathways derive from data and

not theory. That is fine, but differs from the way we came to know anammox, with a chemist staring at thermodynamic tables and positing that anammox was a possible reaction until it was later found. Here is the opposite—great. But are these possible reactions thermodynamically feasible? I have no idea, that is too far from what I know.

Despite that I am excited by the findings in this paper and their ramifications for understanding N cycling, this is but one study and thus too early to “conclude” or to “overturn”.

Specific comments

18 How does it violate? Results needed. The abstract nicely states the context of the study, but it is hard to tell what the authors actually did.

32 Might be worth mentioning that nitrifiers are chemoautotrophs in this general introduction.

64 Most research...

79 Independent of concentrations of nitrate?

80. This is the first sentence of the results. Further from what?

85 are there unrecognized inhibitors of nitrification?

96. Clarity needed. What if the “degree of 15 labeling”? The fraction of the pool that is 15N? Something else? Also, FN could be interpreted as $F \cdot N$; subscript the N.

105 “We could...” Why the conditional?

105 Why “accepted” (and “established” earlier)? Are there disbelievers in isotope pairing math? Is it to distinguish from the new formulations later in the paper? If so say so.

111 See 296

117 Delete clearly

132. Equation could be clearer. Use horizontal line for division. Use single letters for variables (with subscripts). In that case the \times symbol can be deleted.

142. “rather counterintuitively”. This paper is difficult enough to follow. Spell out exactly what you expect, what you found, and why it is counterintuitive.

145. This paragraph is the crux of the authors’ argument, yet is very difficult to follow.

165. “Conclude” is too strong a word for a single lab study. “We suggest that the coupling...”

172. Consider a picture for this point.

185. Overturn is too strong a word. Let’s wait for a few more observations of this phenomenon before we “overturn” our knowledge. This is a single study at a single time.

213. So a big headspace in these vials.

215. Why add ATU? Yes, I see it is in the results, but should be here too

216. That is super high NH₄. I understand why, but given that this paper is sent to a general journal. I think the methods need to be described in a way to link clearly with the hypotheses and to state exactly what the methods are doing and why. E.g. high concentrations of 15NH₄ enable..."

230-235. Split this epically long sentence. On line 232 after conditions state how this approach will achieve this further understanding.

259. That R was used is not that useful; this point can be buried at the end of the paragraph. Instead focus on how the statistical approach enables understanding and inference.

274. This method of providing data seems anachronistic in this age of cloud storage and open data. And who decides what constitutes a reasonable request? Being extra polite? I urge the authors to post the data and code used to analyze them, after publication of course. Conclusions (e.g. line 165) will come only after scrutiny of these data and replication of this work. Code and data are needed for that.

296. "had no effect" is an incorrect interpretation of a null hypothesis significance test. There may very well be an effect, but it is not evident given the variability in the data. $P=0.13$ implies that there might be an effect; we cannot tell. But I cannot make this point strongly enough: $P>0.05$ does not mean the null hypothesis is true. A better way to make the point would be to calculate the parameter estimate and its uncertainty interval. If this interval is smaller than what the authors might say is an effect that matters, then that provides more information saying the false "no effect". I see from the results that the ratios are in fact not that different from one another, good.

Fig 1c. Put error estimates on these points.

Figure 2. In principle, this type a figure is a great way to make the point. In practice, I think this figure needs some work to make these difficult concepts more easy for the reader. First is that the points and fonts are tiny, nearly all focus is on the ribbon. Consider alternative ways of plotting, I am not sure what these would look like. A data visualization specialist might be able to help. One way might be to make a gif where the figure rotates and one can see the ribbon and the other points, but that would be for the supplement

Supplementary equations.

These equations need some work to be clear. First, define all variables here, even if already defined in the text (e.g. D). Give units (or specify if a fraction). What are the rates for D? I think of rates as 1/time. Use a horizontal line for division and not /. Variables should be in italics. I now follow the advice from Edwards and Augur Methe, Methods in Ecology and Evolution 2018 for writing equations. r_a is confusing, it could be r times a , make it r with subscript a (r_a if using LaTeX). These quibbles aside, I really appreciate the details of this math laid out here in this form, and so will the readers.

Reviewer #3 (Remarks to the Author):

In a convincing set of experiments and calculations, the authors show that the production of N₂ from incubations of oxic riverbed slurries cannot be explained by pathways including free nitrite or nitrate pools that can be measured. Instead, it was coupled tightly to the ammonium pool.

While I agree with the authors of this well written manuscript in most points, I disagree in their final conclusion: I do not think that the results call for a novel pathway or novel coupling between

pathways. We can explain the results with known pathways such as nitrifier denitrification or tight coupling, also known as nitrification-coupled denitrification.

Despite this disagreement, I still regard this manuscript as very important and very worth publishing in this journal, as most readers would not be aware that these pathways or such tight coupling could dominate N₂ (or N₂O) production. We mostly think in terms of denitrification or anammox when we think about N₂. Thus, this would be one of the very, very few studies showing N₂ production by other processes.

I have few other comments, provided in an annotated copy.

Nicole Wrage-Mönnig

Reviewers' comments with our replies in italics:

Reviewer #1 (Remarks to the Author):

In “Coupled nitrification and N₂ gas production as a cryptic process in oxic riverbeds,” Ouyang et al present nice experimental data indicating that when ¹⁵N-NH₄⁺ spikes were used to calculate rates of N₂ production, the intermediate, nitrite, did not mix in the sediment pore waters. Thus the basic assumption of the technique that is widely used to obtain N₂ production rates was found to be untrue. While this paper does an excellent experimental job of proving the presence of this problem, the problem, or a very similar problem with ¹⁵N-NO₂⁻, has already been noted and explained in previous papers (see below). I believe that the fact that sediments are composed of particles can explain the author’s data and this should be addressed in the paper. Proposing the activity of a novel player in the N cycle is unnecessary. Overall, after editing, this paper will be a good addition to the literature that will highlight a fundamental problem with a widely used technique.

*Additional data before the details of our reply. There is more to our case than we presented in our original manuscript. We omitted an additional set of treatments from the original manuscript in an effort to try and simplify what was clearly an already complex story. In a parallel set of the same oxic incubations, we added ¹⁵N-tracer as ¹⁵N-nitrite rather than ¹⁵N-ammonia – all with and without ATU etc., (Treatments 5 & 6 in Table 1 below) and measured no significant production of ¹⁵N-N₂ gas. See lines 132-134 and 183-197. We also measured no difference in ²⁹N₂ production from ¹⁵N-ammonia with or without ¹⁴N-nitrite. This is now our major reason for not supporting nitrifier-denitrification or canonical denitrification as sources of N₂ gas in oxic incubations with ¹⁵N-ammonia. Reviewer 1 also raises lots of points in relation to anaerobic nitrate reduction supplying nitrite in oxygen minimum zones (OMZs) and having reviewed the data in support of a coupling between aerobic ammonia oxidation and N₂ gas production in OMZs we have decided to completely remove all mention of OMZs from our manuscript (see **Detailed comment 1**).*

Issue 1: ATU (allylthiourea) may be an inhibitor of long standing, but it was originally tested before we knew about ammonia oxidizing archaea. The authors need to convince the reader that ATU inhibits the organisms that they say it does and not others. I shouldn’t have to figure it out for myself.

1) Martens-Habbena et al 2015 Environmental Microbiology 17: 2261-74. Martens_Habbena et al 2015 indicate that ammonium oxidizing archaea were 97% inhibited at 330 uM ATU. Betaproteobacterial ammonia oxidizing bacteria were inhibited at 10 uM ATU and gammaproteobacterial ammonia oxidizing bacteria were inhibited at 100 uM ATU. The present paper used 2 mM ATU, so everything is inhibited 10x over. However, does ATU inhibit anammox bacteria??

2) Dapena-Mora et al 2007 Enzyme and Microbial Technology 40: 859-865. Not an amazing paper, but it indicates that 1 g/L ATU does not affect anammox bacteria.

3) van de Graaf et al 1996 Microbiology 142: 2187 also says that ATU does not inhibit anammox.

We have completely revised the methods section to improve clarity in light of both reviewer 1’s and 2’s comments. The 2.8mM ATU concentration referred to above is actually the stock solution, whereas the target porewater concentration was ~80µM – we hope this is now clearer in both the main text (line 85) and methods (lines 250-260, 278-280). We used ~80µM as it is in line with the original Hall paper¹ i.e. 86µM for freshwater lake sediments and as it matched what we did before in some of the same riverbed sediments² – so we

knew it would work. Admittedly, at the start of the current work in 2015 we were not aware of Martens-Habbena et al. 2015 paper. Clearly Martens-Habbena et al. show that marine strains of ammonia oxidizing archaea (AOA) are only inhibited at higher concentrations of ATU (330 μM) and not the “standard”, freshwater 80-100 μM applied here; though they did confirm inhibition of both β - and γ -proteobacterial ammonia oxidising bacteria (AOB). We did know about the van de Graaf et al (1996) paper and the lack of effect of ATU on anammox and also the Jensen et al (2007)³ paper demonstrating no effect on denitrification and anammox (now cited on line 86). The important point here though is that we added ATU to establish a link between aerobic ammonia oxidation and subsequent N_2 gas production and clearly from the data in Fig. 1 and Table 1, adding ATU at $\sim 80\mu\text{M}$ had the desired effect. We have added a caveat to the methods saying that higher concentrations may be required in other settings citing Mattens-Habbena, line 255 but do not want to open this in the main text as it is tangential to the main story.

We do have molecular data for all 12 riverbed sediments and, briefly, AOA and AOB are abundant (qPCR, *amoA* – Fig. 1. below) and the AOA *amoA* gene sequences are closely related to *Nitrosopumilus maritimus* and *Nitrososphaera viennensis*. Despite the presence of AOA, ATU inhibited the aerobic oxidation of ammonia in these freshwater sediments. We have included a note in the Methods and cited additional papers on lines 87 and 255.

Fig. 1. Simple presentation of AOB vs. AOA abundances (*amoA* copies g^{-1} sediment). Yellow and blue are gravel and sand dominated riverbeds, respectively.

Issue 2: Similar phenomena in oceanic oxygen deficient zones using ^{15}N - NO_2^- have published previously in three papers. These papers should all be cited. Both De Brabandere et al 2014 Environmental Microbiology 16: 3041-3054. and Chang et al 2014 Limnology and Oceanography 59: 1267-74 find an excess of $^{29}\text{N}_2$ produced in enriched ^{15}N experiments with nitrite compared to experiments with ^{15}N - NH_4^+ .

Basically, as the authors noted in the present paper, in these two oxygen minimum zone papers, the isotopic composition of labeled N_2 cannot be explained by the binominal distribution when anammox is taken into account. In Chang and De Brabandere, there is too much $^{29}\text{N}_2$ for the amount of denitrification that should be present. Both papers explain this data by suggesting an internal nitrite pool for denitrifiers. Which is to say, that the nitrite used by denitrifiers is not mixing with the spiked surrounding water. Exactly as seen in the current paper in pore waters. Nicholls et al 2007 also see a similar phenomena. (co-authored by authors of the current paper but not cited in the current paper) Internal nitrite pools have been indicated in denitrifiers in a sulfidic fjord by Jensen et al 2009 Marine Chemistry 113: 102-113.

Fuchsman et al 2018 Deep Sea Research II 156: 137-147 examined natural stable isotopes of N_2 and nitrite and nitrate. The fractionation factor from (nitrate and nitrite) to N_2 gas changed with depth in the oxygen minimum zone. Only two hypotheses could explain this phenomenon: an internal nitrite pool in denitrifying bacteria or an internal nitrite pool inside of sinking particles. Mathematically the results of these two situations are similar.

We are aware of the earlier studies implying internal nitrite pools. An important distinction, however, is that the experimental studies mentioned employ (near-)anoxic incubations and suggest an internal pool within a group of anaerobes (denitrifiers) already known to carry out the entire pathway in question (the stepwise reduction of nitrate to N₂), whereas explaining our observations with existing models requires two types of organisms with opposing oxygen requirements (aerobic ammonium oxidizers and anaerobic N₂-producers). Thus, while the earlier observations can be explained by an intracellular nitrite pool with limited exchange to the outside, such an explanation is not possible in our case.

Furthermore, there is a lot here in relation to anaerobic nitrate reduction supplying nitrite in OMZ waters. We looked again at the evidence for a coupling between aerobic ammonia oxidation and N₂ gas production at the margins of OMZs. While the earlier papers may have suggested a link (Kuypers et al. 2007 “indicating that aerobic oxidation of ammonium, rather than nitrate reduction, is the source of nitrite for anammox...”; Lam et al., 2009 – note Lam et al., 2007 was a wrong click in EndNote) there was little concrete evidence. In hindsight, perhaps we should not have mentioned OMZs; though there will be some who consider the coupling a possibility. Further, plotting the data from Kavelage et al. (2013) shows a nice correlation (r=0.83) between nitrate reduction to nitrite and anammox but no such relationship between aerobic ammonia oxidation and anammox (r=0.02). In light of this, we have removed all mention of OMZs from the text and now, also in reply to reviewer 3, state our focus on sediments as major sources of N₂ on Earth. Lines 62-64.

Particles have diffusive boundary layers that separate their insides and outsides (Ploug et al 1997 Aquatic Microbial Ecology 13: 285-294) and often have elevated nutrients inside the particles (Simon et al 2002 Aquatic Microbial Ecology 28: 174-211). Wilson et al 2014 Deep Sea Research I 114: 47-55 has shown elevated nitrite in oxic particles in the ocean.

Yes particles have diffusive boundary layers and yes the data in Simon et al., show elevated concentrations for nitrate and ammonia on the inside of particles but, just to be clear, Wilson, who we contacted, said “However we did not see any increase in NO₂-” i.e., they did not measure elevated nitrite in oxic particles. Besides, the take home message in the Ploug et al. (1997) is that anoxia (in highly organic i.e. 100%, lab-made aggregates) is an “ephemeral phenomenon” that they only found in 2 out of 8 aggregates and that “...would limit anoxic conditions to occurring only over a few hours, depending on the size of the aggregates.” Further, they say “The volumetric oxygen respiration rate around an anoxic center in a 1.4 mm large aggregate (Fig. 2) was calculated as 22.1 nmol O₂ mm⁻³ h⁻¹”. We typically measure oxygen respiration at ~ 0.1 nmol O₂ mm⁻³ h⁻¹ (or about 187 nmol O₂ g⁻¹ h⁻¹). They then model anoxia as a function of oxygen consumption in different size spheres and show that at 250 μM oxygen in the surrounding water (i.e., very similar to our conditions) respiration would need to be 3-5 nmol O₂ mm⁻³ h⁻¹ to make a 2mm aggregate anoxic. Yes, our largest gravels particles (in the Exetainers) may have been 2mm but they are principally inorganic rock fragments (0.34% organic C dry wt) with thin layers of organic biofilm which, if 100 μm thick, would require respiration of 1400-4200 nmol O₂ mm⁻³ h⁻¹ to become anoxic.

In ocean oxygen minimum zones, denitrifiers are enriched on particles (Ganesh et al 2015 ISME 9: 2682-2696 and Fuchsman et al 2017 Frontiers in Microbiology 8: 2384.). It is likely that internal nitrite pools in particles can explain this entire phenomena including data from Chang et al 2014 and De Brabandere et al 2014.

See reply to Issue 3 where the reviewer returns to internal nitrite pools.

Issue 3. The data in the current paper can be explained by particles. The current paper takes place in sediments. Sediments are composed of particles. We already know that the

fact that sediments are composed of particles greatly affects N cycling there. Particles are the explanation why the natural fractionation of N₂ production in sediments is close to zero. Nitrate is completely consumed in the particles. *Implicit in this accepted argument is the idea that the inside of particles and the pore waters are not exchanging quickly. See: Brandes and Devol 1997 *Geochimica et Cosmochimica Acta* 61: 1793-1801 Lehmann et al 2007 *Geochimica et Cosmochimica Acta* 71: 5384-5404.

**For this to explain the current data, I believe that spiked ammonium would have to diffuse into particles but nitrite would not diffuse back out. It becomes part of an internal nitrite pool. This pool could be inside the particle or inside an organism. The experiments in the current paper were 8 hours long—pretty short. Probably a longer incubation would have more exchange. Ammonium is a precious item that organisms want. Particles also explain anaerobic processes under oxygen. This entire phenomenon is related to particles and particles need to be addressed in the paper. See Steif et al 2016 *Frontiers in Microbiology* 7: 98. Ploug and Bergkvist 2015 *Marine Chemistry* 176: 142-149. Bianchi et al 2018 *Nature Geoscience* 11: 263.

*The reviewer raises lots of points under **Issue 3**. The nature of the particles was addressed in Issue 2. Thus we start by addressing their summing up comments* and **, above: "... For this to explain the current data, I believe that spiked ammonium would have to diffuse into particles but nitrite would not diffuse back out. It becomes part of an internal nitrite pool."*

Our point is that nitrite does diffuse back out. We added ¹⁵NH₄⁺ to fully oxic incubations, we saw pretty much immediate production of ²⁹N₂ and ³⁰N₂ and recovered both ¹⁵NO₂⁻ and ¹⁵NO₃⁻ from the surrounding porewater (Fig. 1a, b). Our oxic incubations are net sources of ¹⁵NO₃⁻ from ¹⁵NH₄⁺ and are in stark contrast to the mm-sized algal aggregates described above as net sinks for nitrate (Stief et al 2016). Those aggregates are in the same category of high respiration rates as the ones discussed in Issue 2 above, and are hence very different from the sediment particles in our study.

It is this combined presence of ¹⁵NH₄⁺ and ¹⁵NO_x⁻ (¹⁵NO₂⁻ and ¹⁵NO₃⁻) that give us our two ¹⁵N porewater pools F_A and F_N (Table 2) that should both – from the accepted point of view – influence the ratio of ²⁹N₂ and ³⁰N₂ produced (Fig. 2a) but they do not (Fig. 3b,c and Fig. 4). We of course agree that a cryptic intermediate pool and/or process could be inside a particle and/or an organism (that is the point of our paper) but we do not agree that it is as simple as an internal nitrite pool for the very reason that we see ¹⁵NO₂⁻ from ¹⁵NH₄⁺ "on the outside" in the porewater.

As a point of detail, the incubations were 12 hours long, not 8, with steady production of ¹⁵N₂ over the first 9-10 hours (Fig. 1). We don't see why this is "pretty short" from a diffusion perspective? We were working with samples of gravel and sand dominated sediments with particles in the incubations of up to 2mm – though these would be largely inorganic with thinner organic biofilms (as above). If we take 2mm as the upper size limit and porosity of either 0.45 or 0.9, then nitrate (Li & Gregory, 1974), nitrite, ammonia or oxygen would diffuse across 2mm in ~ 0.25h to 1.2 h at 12°C. For a 100µm thick organic biofilm (as above) this would be ~0.04 minutes. For "the inside of particles and the pore waters are not [to be] exchanging quickly" cannot be down to diffusion, there would need to be controlled, limited exchange.

Ammonia is of course a rare commodity in the ocean but it accumulates in sediments and in these riverbeds can range in situ from 10s to 1000s of µM – hence, spiking the porewater to ~390 µM ¹⁵NH₄⁺ only labelled ~50% of the ammonia pool. As we describe above, oxygen

consumption would need to be closer to 3-5 nmol O₂ mm⁻³ h⁻¹ for the core of a 2mm organic aggregate to go anoxic in 250µM oxygen porewater, whereas we routinely only measure 0.1 nmol O₂ mm⁻³ h⁻¹ with about 0.3% organic carbon by dry wt.

Finally, the reviewer states very firmly that “Nitrate is completely consumed in the particles” but we would just like to point out that there is no experimental evidence for this rather it is implied from modelling gradients in algal aggregates in low nitrate waters (Klawon et al. 2020)⁴.

Issue 4. Given that the data in the paper can be explained by denitrification in sediment particles, which we already know happens, it seems like invoking a new player (or coupling) in the N cycle is unnecessary.

Quite simply, if ¹⁵NO₂⁻ and ¹⁵NO₃⁻ can diffuse “out” (Fig. 1b) from where ever ¹⁵NH₄⁺ is being oxidised, then ~24µM ¹⁵NO₂⁻ added to the porewater can diffuse “in”, and if the “aggregates” can go anoxic as the reviewer suggests – bearing in mind that would require far higher respiration than we measured, then we would expect production of ¹⁵N₂ from ¹⁵NO₂⁻ – but we did not measure that (Table 1, main text and below). If, however, you make exactly the same sediment slurries anoxic and add ¹⁵NO₂⁻ or ¹⁵NO₃⁻ then you will measure denitrification, just as we have shown before for the rivers Nadder, Avon (west branch), Ebble, Wyle and Avon (east branch) in² and for the Hammer, Medway, Marden, Nadder, Stour I, Stour II and Lambourn in⁵ (i.e. 10 of the 12 rivers visited in this study, see Supplementary Table 1).

What you measure in oxic sediments incubated with ¹⁵NH₄⁺ is not the same as what you measure in anoxic sediments incubated with ¹⁵NO₂⁻. Admittedly, we are not sure what process is making the N₂ at the moment but it is not as simple as anammox and/or denitrification.

Table 1. Overall summary of the production of ¹⁵N₂ over 12 hours from oxic sediments incubated with treatments 1 to 6, for the first four rivers, and treatments 1 and 2 for the second twelve rivers. Only with ¹⁵NH₄⁺ and without ATU, and either with or without ¹⁴NO₂⁻, do we see any significant production of ¹⁵N₂ gas.

Treatment	Rivers (replicates)	Total ¹⁵ N-N ₂ (nmol N g ⁻¹ h ⁻¹)	s.e.	Lower 95% C.I.	Upper 95% C.I.
1 , ¹⁵ NH ₄ ⁺	4 (5)	1.855	0.326	1.078	2.631
2 , ¹⁵ NH ₄ ⁺ + ATU	4 (5)	0.110	0.337	-0.667	0.886
3 , ¹⁵ NH ₄ ⁺ + ¹⁴ NO ₂ ⁻	4 (5)	1.941	0.326	1.165	2.717
4 , ¹⁵ NH ₄ ⁺ + ¹⁴ NO ₂ ⁻ + ATU	4 (5)	0.152	0.337	-0.625	0.929
5 , ¹⁴ NH ₄ ⁺ + ¹⁵ NO ₂ ⁻	4 (5)	0.279	0.326	-0.497	1.055
6 , ¹⁴ NH ₄ ⁺ + ¹⁵ NO ₂ ⁻ + ATU	4 (5)	0.314	0.326	-0.462	1.091
1 , ¹⁵ NH ₄ ⁺	12 (5)	1.465	0.176	1.091	1.839
2 , ¹⁵ NH ₄ ⁺ + ATU	12 (5)	0.129	0.178	-0.249	0.506

Detailed comments:

1, Line 51: “Similarly, aerobic nitrification can fuel both anaerobic denitrification⁹ and anammox¹⁰ at the margins of the vast oceanic oxygen minimum zones” Here ref 9 is Ward et al 2009—that paper has absolutely nothing to do with aerobic nitrification fueling denitrification. I just reread to be sure.

In reply to detailed comments 1 to 4 we looked again at the evidence for a coupling between aerobic ammonia oxidation and N₂ gas production at the margins of OMZs. While the earlier papers may have suggested a link (Kuypers et al. 2007 “indicating that aerobic oxidation of ammonium, rather than nitrate reduction, is the source of nitrite for anammox...”; Lam et al., 2009 – note Lam 2007 was a wrong click in EndNote) there was little concrete evidence. Further, plotting the data from Kalvelage et al. (2013) shows a nice correlation (r=0.83) between nitrate reduction to nitrite and anammox but no such relationship between aerobic ammonia oxidation and anammox (r=0.02). In light of this, we have removed all mention of OMZs from the text and now, also in reply to reviewer 3, state our focus on sediments as major sources of N₂ on Earth. Lines 62-64.

2, Reference 10 is Lam et al 2007, which is about aerobic nitrification fueling anammox (specifically) in the Black Sea. The Black Sea is fundamentally different from oceanic oxygen minimum zones in that it has a large flux of ammonium coming from its underlying sulfide zone. If you would like to see these fluxes quantified see Fuchsman et al 2008. On the other hand, in the giant oceanic oxygen minimum zone, ammonium is below detection (<10 nM) in the most of the oxygen minimum zone. See both Widner et al 2018s for ammonium data in the ETNP and ETSP.

See above, reply to detailed comment 1.

3, Bristow et al 2016 indicates that the Km for oxygen for aerobic ammonium oxidation is higher than the Km for nitrite oxidation. Rather than aerobic ammonium oxidation fueling denitrification/anammox in the oceanic oxygen minimum zones, nitrate reduction coupled to nitrite oxidation is a key process using the small concentrations of oxygen. Also note that, unlike the Black Sea, in OMZs nitrate concentrations are >20 uM from ambient seawater. aka oxidized N is not a limiting factor for denitrification.

See above, reply to detailed comment 1.

4, Please delete. Also this means that the authors have to reword that entire paragraph.

See above, reply to detailed comment 1 and the rewrite of lines 52-64.

5, Line 96: This paper uses a lot of math. It might help the reader to follow if instead FA from fraction of ammonium, you used FNH₃ and FNO_x because FA could also be fraction anammox. It is a simple change that could make things easier to follow.

Reviewer 2 also asked us to revise the mathematical notation, symbols and equations. We have revised all the equations in both the text and SI. We use ‘x’ to denote multiplication and have revised F_N and F_A with subscripts throughout. See line 95-113, 302-305 and the Supplement. For convention, we propose to retain “ra” for the relative contribution from anammox to the overall production of N₂, as proposed by Risgaard-Petersen et al. (2003), as it is now part of this field’s literature and, to avoid any possible confusion with “r x a”, we have retained x to indicate multiplication in all the equations.

6, Perhaps Table 1 should include the oxygen concentrations for each experiment.

We did not measure oxygen concentrations in parallel incubations for all 12 rivers as that would have been quite simply impossible. However, we do know from our former work in

many of the riverbeds revisited again here² and our extensive studies across the southern UK⁶ their respiration rates and that oxygen will not run out. We now state in the methods, lines 236-244, that “Oxic slurries were prepared by adding approximately 3 g sediment (~0.75 ml of porewater) and 2.7 ml air-saturated synthetic river water into 12 ml gas-tight vials (Exetainer, Labco), leaving an approximate 6 ml headspace of air which is equivalent to ~58 $\mu\text{mol O}_2$ per prepared vial. We know from previous incubations with similar sediments from 28 rivers⁶ respiration rates to be ~187 $\text{nmol O}_2 \text{ g}^{-1} \text{ h}^{-1}$, on average (± 64.3 , 95%, C.I.), that would consume ~12% of the total oxygen during a 12h incubation. In addition, we also checked oxygen over time using a microelectrode (50 μm , Unisense) in parallel sets of scaled-up slurries (120 mL and same ratio of sediment to water to headspace) for two rivers and found comparatively little consumption as before² (Supplementary Figure 2).”

7, Line 188: Here we go again. The Black Sea is not like the margins of oxygen minimum zones due to the large flux of ammonium into the Black Sea suboxic zone. Both references 10 and 30 are about the Black Sea. The Kalvelage et al 2011 reference is more legitimate. However, Kalvelage assumes rather than shows that aerobic ammonium oxidation and anammox are linked. If you want to say this, you should really be citing Lam et al 2009 PNAS. That is where this idea is originally coming from. In that paper it comes from a calculation based on the balance of rates. However, back in the day of the Lam paper, the rubber stoppers were infecting denitrification rate measurements with oxygen and inhibiting them, so that calculation is very probably incorrect.

See above, reply to detailed comment 1. All reference to OMZs removed.

8, If you look at models by Zakem et al 2020 ISME 14: 288-301—like Fig 3—you will see that aerobic ammonium oxidation to nitrite is not a key player in the OMZ system or its edge. You don't need to cite oxygen minimum zones for this process to be important. Sediments are quite important by themselves.

See above, reply to detailed comment 1. All reference to OMZs removed.

Reviewer #2 (Remarks to the Author):

This paper uses isotope pairing methods on oxic riverine sediments in a lab to deduce that there is a cryptic intermediate pool of oxidized N that is not part of the measurable porewater pool. If true, this is a tremendously interesting finding. My criticisms of the paper have nearly nothing to do with the “if true” part. This is high-risk, high-reward science and as such I support it fully. Methods and modeling are as good as they can be to address this question in this paper. My criticisms are with manuscript presentation and explanation of how the authors link that data and mathematical framework to new understanding. Thus my comments below are meant to improve the manuscript and not to argue against its publication.

1, I struggled to follow along with the paper from the methods the data and the mathematics. There are two reasons for this. 1. This work is inherently highly technical and never is going to be easy to understand. Despite 25 y of N isotope experience and a recent foray into isotope pairing in my lab, this paper took a lot of work to review. The authors may wish that I were smarter, and I would agree with them. But I am not, so 2. I think that there is a lot of room for clarifying the paper to enable readers to follow along with the reasoning. This clarification can include: mathematical notation, explanation of methods, figures. The storyline of the paper is very much falls along how the authors discovered it: data followed by modeling to see what is possible and then a conclusion. Would putting the modeling first

followed by data that supports (or not) the models work better? I have a lot of comments below.

See combined reply to points 1 & 2 below.

2, Consider some sort of diagram that outlines the methods and how they lead to a conclusion. A flowchart may work, but this is not to be prescriptive. But I think at any way to lead the reader from the methods to results to understanding would be a big help.

In reply to points 1 and 2. We have completely overhauled the paper, rewriting to simplify large sections of the text, methods and revising the mathematical notation. We have also increased the number of figures from 2 to 4 in order to breakdown and simplify the presentation of the data which we hope helps to map the data onto the narrative and methods. We now also include a schematic diagram (Fig. 2) to illustrate the accepted (2a) framework for how substrate pools and reactions interact to generate N_2 gas and our proposed alternative (2b). We now also include Broda's 1977 thermodynamic predictions for the complete aerobic oxidation of ammonia to N_2 (equation/reaction 5, line 51) which provides a theoretical framework for our data and – as we now more explicitly argue at the end of our paper, lines 198-206, – actually offers the simplest explanation for our data (more details below).

3, This paper lives and dies on it methods. The authors have done new techniques to putatively show new N pathways. But the methods section is written very tersely for few specialists who do these methods. I think that there should be more text linking what the authors want to know with how the method addresses this point. I admit to having trouble making these points myself despite having just starting using isotope pairing methods in my lab to attempt to measure denitrification in oxic aquifers. The Results section has a lot of this text and yes, it covers some of the methods there, great. But I would spell out these details in both places.

This is a perfectly valid criticism of the methods section from a non-specialist's perspective and we have now revised them extensively. Including: a detailed explanation of the availability of oxygen throughout the 12h incubations (in reply to reviewer 1) on lines 236-244; an explanation of what is being traced with ^{15}N -ammonia; why we enriched the ammonia pool by $\sim 400\mu M$; and clearer phrasing of the use of allylthiourea on lines 251-255. We now also include data for two additional treatments (5&6, Table 1) that were omitted from the original manuscript in an effort to simplify the story. Here the incubations were exactly as described for ^{15}N -ammonia but instead we added ^{15}N -nitrite and state which reactions they were screening for – see detailed reply to reviewer 1 and lines 284 to 290. We now also describe explicitly why and how we manipulated the fraction of ^{15}N labelling in the NO_2^- pool to drive changes in the ratio of $^{29}N_2$ to $^{30}N_2$ produced (lines 271-283) and equations to show how the two key components F_A and F_N are calculated (lines 302-308). See other details below in relation to specific comments.

4, The arguments for what is actually happening to drive these patterns (starting on line 171) are critical parts of the paper. But I note with interest how the discovery of these pathways derive from data and not theory. That is fine, but differs from the way we came to know anammox, with a chemist staring at thermodynamic tables and positing that anammox was a possible reaction until it was later found. Here is the opposite—great. But are these possible reactions thermodynamically feasible? I have no idea, that is too far from what I know.

The reviewer raises an interesting point that prompted us to revisit Broda's 1977⁷ thermodynamic ponderings that, along with Richards observations (1965)⁸, drove the original quest for anammox (anaerobic ammonium oxidation). Not only did Broda propose the

potential for anammox through either reaction 1 or 2 (below) in nature, where 2 was later proven to be the anammox reaction⁹, he also postulated the complete oxidation of ammonia with oxygen to N₂ gas through reaction 3. This aerobic oxidation of ammonia to N₂ has rough thermodynamic equivalence to anammox (2) and complete aerobic nitrification to NO₃⁻, reaction 4. We have now included this in our introduction and discussion. So yes, the reaction is thermodynamically feasible and it actually offers the simplest explanation for our data. See lines 48-52 of the introduction and lines 198-206 of the discussion.

Despite that I am excited by the findings in this paper and their ramifications for understanding N cycling, this is but one study and thus too early to “conclude” or to “overturn”.

We appreciate the reviewer’s excitement for our work. Whereas this may only be one study but so were the original studies on anammox, comammox and many, many others and few, if any, presented findings for 12 genuinely independent field sites, with many basing their conclusions on pseudoreplication. We have, however, toned down “overturn” to “challenges” and “conclude” to “propose”: “We propose that the coupling between ammonia oxidation and N₂ gas production in oxic, permeable riverbed sediments involves a cryptic intermediate pool...”. It is certainly cryptic from the perspective of the currently accepted mathematical framework e.g. that of Song¹⁰ but also many others too. See lines 174-178.

Specific comments

18. How does it violate? Results needed. The abstract nicely states the context of the study, but it is hard to tell what the authors actually did.

We have revised the abstract throughout and now include a key “simple” fact identified through the revisions to points 1 and 2 above. We do however want to keep any technical jargon to a minimum for the more general audience we are targeting.

32. Might be worth mentioning that nitrifiers are chemoautotrophs in this general introduction.

Done, see line 32.

64. Most research...

yes, more concise. Done, see line 65.

79. Independent of concentrations of nitrate?

No, it is not the concentration per se, it is simply the fact that the pattern of ¹⁵N-labelling in the porewater NO_x⁻ pool had no discernible influence on the ¹⁵N-labelling of the N₂ gas produced – that you would expect according to the accepted mathematical framework common to this type of ¹⁵N-work in sediments. Concentration may affect the rate of reaction but that would apply equally to both ¹⁴N and ¹⁵N so wouldn’t affect the “patterns”.

80. This is the first sentence of the results. Further from what?

It was meant to be further to our former work published in 2016 (cited² and described in the section above) on the coupling between aerobic nitrification and putatively anaerobic N₂ gas production in oxic riverbeds. We have rephrased to make it more explicit. See line 79-81.

85. are there unrecognized inhibitors of nitrification?

See detailed reply to reviewer's 1 first point and main text (line 85) and methods (lines 250-255, 274-279).

96. Clarity needed. What if the “degree of 15 labeling”? The fraction of the pool that is 15N? Something else? Also, FN could be interpreted as F*N; subscript the N.

We have rewritten this entire section as suggested, added equations to the methods (10 & 11) to show how F_N and F_A are calculated, shortened sentences, and revised all the equations in both the text and supplement. We use 'x' to denote multiplication and have revised F_N and F_A with subscripts throughout. See line 94 onwards, 302-305 and the Supplement.

105. “We could...” Why the conditional?

Revised, see line 109 and other extensive revisions either side of this aimed at clarifying the link between our methods and results.

105. Why “accepted” (and “established” earlier)? Are there disbelievers in isotope pairing math? Is it to distinguish from the new formulations later in the paper? If so say so.

The latter - first occasion is now “The published and accepted mathematical...” and the second “accepted mathematical”. See the completely revised lines 94-108, 109-139 and 141 to 148.

111 See 296

For 111 and point 296 below. We have completely revised both the way we present and analyse the data for the ratios of ²⁹N₂ to ³⁰N₂ i.e. R. We have now calculated averages and confidence intervals for both our measured R values and the predicted R values (equations 6 and 7, respectively) for both denitrification and anammox. See lines 109-139 and Table 2. In the new Fig. 3 we use boxplots to show the overall spread in the data, plotting the measured and predicted values next to each other to show the discrepancy between the two. In the text, the statement “diluting F_N had no discernible effect on the values for R...” is then followed by the averages with their 95% CI and reference to Table 2. The figure caption that contained text to the same effect i.e., “see 296” has been completely revised – along with the figure – to form the new figure 4.

117. Delete clearly.

Done.

132. Equation could be clearer. Use horizontal line for division. Use single letters for variables (with subscripts). In that case the \times symbol can be deleted.

We have revised equations 6 and 7 (now 8 and 9) in the main text (see lines 146-153) and equations 1 to 14 in the supplement. For convention, we propose to retain “ra” for the relative contribution from anammox to the overall production of N₂, as proposed by Risgaard-Petersen et al. (2003), as it is now part of this field's literature and, to avoid any possible confusion with “r x a”, we have retained 'x' to indicate multiplication in all equations.

142. ‘rather counterintuitively’. This paper is difficult enough to follow. Spell out exactly what you expect, what you found, and why it is counterintuitive.

We really do not want our paper to be difficult to follow at all. We have removed “rather counterintuitively” and extensively revised the text both leading to and following on from this section and hope that helps. Lines 154-156 and the revisions either side for context.

145. This paragraph is the crux of the authors’ argument, yet is very difficult to follow.

Addressing point 111 above led to us revising all of the figures which, in turn, enables us to first introduce the reader to a clearer, simpler presentation of the discrepancies in our data i.e., the simple comparison of our measured R values and those predicted by the accepted mathematical framework. This then enabled us to simplify the crux of our argument that revolves around the now fully revised 3D figure (see point below, now Fig. 4). We hope that these simplifications make our argument easier to follow. Lines 157-172, revised Fig. 4 and new Fig. 2.

165. “Conclude” is too strong a word for a single lab study. “We suggest that the coupling...”

As above “We propose”. Line 174 onwards.

172. Consider a picture for this point.

We now include the schematic Fig. 2 showing both the accepted scenarios and our proposed cryptic scenario which we refer to throughout the text to help guide the reader.

185. Overturn is too strong a word. Let’s wait for a few more observations of this phenomenon before we “overturn” our knowledge. This is a single study at a single time.

As above, replaced “overturn” with “challenge” on line 207 but also see the extensive revisions either side for context.

213. So a big headspace in these vials.

We have completely revised the methods section as described above for reviewer 1 and now describe what the 6 mL headspace is equivalent to as an amount of oxygen and how that compares to our measured rates of respiration for these gravel/sand sediments. Line 236-244.

215. Why add ATU? Yes, I see it is in the results, but should be here too.

As above, clarified, see line 250-262.

216. That is super high NH₄. I understand why, but given that this paper is sent to a general journal. I think the methods need to be described in a way to link clearly with the hypotheses and to state exactly what the methods are doing and why. E.g. high concentrations of 15NH₄ enable...”

As above and see lines 248-250 for an explanation in relation to what it is we were measuring. It is not actually that high and we only labelled about 50% of the ammonia pool.

230-235. Split this epically long sentence. On line 232 after conditions state how this approach will achieve this further understanding.

The whole methods section has been completely rewritten in light of this and the previous comments. See lines 245-290.

259. That R was used is not that useful; this point can be buried at the end of the paragraph. Instead focus on how the statistical approach enables understanding and inference.

We have moved R and other package details to the end of this section but are not sure what else the reviewer wants us to include in what is just a fairly routine description of our statistics? Further details are provided in both the main text and supplemental tables. See lines 318-336.

274. This method of providing data seems anachronistic in this age of cloud storage and open data. And who decides what constitutes a reasonable request? Being extra polite? I urge the authors to post the data and code used to analyze them, after publication of course. Conclusions (e.g. line 165) will come only after scrutiny of these data and replication of this work. Code and data are needed for that.

This is just standard text from the Nature guide to authors. We can provide the data presented in the figures if we get published.

296. “had no effect” is an incorrect interpretation of a null hypothesis significance test. There may very well be an effect, but it is not evident given the variability in the data. $P=0.13$ implies that there might be an effect; we cannot tell. But I cannot make this point strongly enough: $P>0.05$ does not mean the null hypothesis is true. A better way to make the point would be to calculate the parameter estimate and its uncertainty interval. If this interval is smaller than what the authors might say is an effect that matters, then that provides more information saying the false “no effect”. I see from the results that the ratios are in fact not that different from one another, good.

See detailed reply to point 111 above and the revised versions of all the figures.

Fig 1c. Put error estimates on these points.

See detailed reply to point 111 above and the revised versions of all the figures.

Figure 2. In principle, this type a figure is a great way to make the point. In practice, I think this figure needs some work to make these difficult concepts more easy for the reader. First is that the points and fonts are tiny, nearly all focus is on the ribbon. Consider alternative ways of plotting, I am not sure what these would look like. A data visualization specialist might be able to help. One way might be to make a gif where the figure rotates and one can see the ribbon and the other points, but that would be for the supplement.

We have completely revised all of the figures. Specifically in relation to the 3D “ribbon”, which is now Fig. 4, we have: reduced the number of data points to just the overall average estimates for R; removed the too busy legend; increased the size of the symbols and axes labels; truncated the ribbon by reducing R from -2 to -1 i.e. the x-axis; and also provided two zoomed-in views from different angles. The senior author, Trimmer, does not have access to any data visualisation specialists at Queen Mary – or anywhere else as all spending at QM has been frozen – and this is the best that can be done with the resources available. I think it is quite clear that our measured values fall near to each other and the gap where there are no solutions, while those predicted simply for denitrification do not.

Supplementary equations.

These equations need some work to be clear. First, define all variables here, even if already defined in the text (e.g. D). Give units (or specify if a fraction). What are the rates for D? I think of rates as 1/time. Use a horizontal line for division and not /. Variables should be in

italics. I now follow the advice from Edwards and Augur Methe, *Methods in Ecology and Evolution* 2018 for writing equations. ra is confusing, it could be r times a , make it r with subscript a (r_a if using LaTeX). These quibbles aside, I really appreciate the details of this math laid out here in this form, and so will the readers.

As we say above, we have completely revised the equations in both the main text and supplement. We have added rate units ($\text{nmol N g}^{-1} \text{ dry sediment h}^{-1}$) for both denitrification and anammox (though any unit would do here) and also defined both F_N and F_A as the fraction of either porewater pool labelled with ^{15}N and now show how each is calculated in the methods, lines 302-305. For convention, we propose to retain “ ra ” for the relative contribution from anammox to the overall production of N_2 , as proposed by Risgaard-Petersen et al. (2003), as it is now part of this field’s literature and, to avoid any possible confusion with “ $r \times a$ ”, we have retained x to indicate multiplication in all the equations.

Reviewer #3 (Remarks to the Author):

In a convincing set of experiments and calculations, the authors show that the production of N_2 from incubations of oxic riverbed slurries cannot be explained by pathways including free nitrite or nitrate pools that can be measured. Instead, it was coupled tightly to the ammonium pool.

While I agree with the authors of this well written manuscript in most points, I disagree in their final conclusion: I do not think that the results call for a novel pathway or novel coupling between pathways. We can explain the results with known pathways such as nitrifier denitrification or tight coupling, also known as nitrification-coupled denitrification.

Despite this disagreement, I still regard this manuscript as very important and very worth publishing in this journal, as most readers would not be aware that these pathways or such tight coupling could dominate N_2 (or N_2O) production. We mostly think in terms of denitrification or anammox when we think about N_2 . Thus, this would be one of the very, very few studies showing N_2 production by other processes.

Both reviewers 1 and 3 say there is no need to call for a novel pathway. We would like to start by reiterating that we only ever posed it as a possibility and left it open i.e., “Whether it transpires that our cryptic coupling is mediated by a novel organism or, as of yet, masked combination of known players in the nitrogen cycle remains to be resolved.” We also included complete nitrifier-denitrification as a possibility, describing the need for nitrosocyanin as a plausible substitute to canonical N_2O -reductase to enable complete nitrifier-denitrification to N_2 gas¹¹.

However, there is more to our case than we presented in our original manuscript. We omitted an additional set of treatments from the original manuscript in an effort to try and simplify what was clearly an already complex story. In a parallel set of the same oxic incubations, we added ^{15}N -tracer as ^{15}N -nitrite rather than ^{15}N -ammonia – all with and without ATU etc., (Treatments 5 & 6 and Table 1) and measured no significant production of ^{15}N - N_2 gas. See lines 131-134 and 183-197. We also measured no difference in $^{29}\text{N}_2$ production from ^{15}N -ammonia with or without ^{14}N -nitrite (Supplementary Table 2). This is now our major reason for not supporting nitrifier-denitrification or canonical denitrification as sources of N_2 gas from ^{15}N -ammonia. As we do not wish to get into an argument over complete nitrifier-denitrification to N_2 we emailed the reviewer (as they had signed their review) for clarification on this point.

The reviewer's claim below (**point for line 62**) that "we also know that nitrifiers... reduce nitrite to N_2O and **also N_2** in nitrifier denitrification" is not actually known for N_2 . Instead, and as we stated, complete nitrifier-denitrification to N_2 would require nitrifiers to have nitrosocyanin as a plausible substitute to canonical N_2O -reductase to enable them to completely reduce nitrite to N_2 gas¹¹. As for reviewer 2, we do now formally introduce the complete aerobic oxidation of ammonia to N_2 as a further possibility, as proposed by Broda (equation 5) and just as Broda originally proposed for anammox. See lines 48-51, 183-197.

In the revised final section "**Internal NO_x^- cycling or a novel pathway or organism**" we now discuss more fully these two scenarios along with the known potential coupling between nitrifiers and anammox bacteria as in CANON reactors. This section now comes to the open conclusion "Regardless of the actual pathway that produces the N_2 gas (Fig. 2b), an isolated cryptic intermediate pool has to have the same ^{15}N -labelling of the ammonia pool ($F_{N_{cry}} = F_A$). As a consequence of this equality, we can no longer distinguish between sources of N_2 gas, be it complete nitrifier-denitrification, canonical denitrification, anammox or complete aerobic ammonia oxidation, as they would all produce $^{29}N_2$ and $^{30}N_2$ at the same ratio (Fig. 2b where R is equal for each process).

Our observations challenge the current understanding of a key coupling in the nitrogen cycle in permeable, oxic riverbed sediments that may also apply to other biomes where the oxidation of ammonia is tightly coupled to the production of N_2 gas, such as continental shelf-sediments^{12,13} and groundwater aquifers¹⁴. Whether it transpires that our cryptic coupling is mediated by a novel organism or, as of yet, masked combination of known players in the nitrogen cycle remains to be resolved." See lines 201-212.

Hence to simply state "We can explain the results with known pathways such as nitrifier denitrification or tight coupling, also known as nitrification-coupled denitrification" is not quite true for nitrifier-denitrification nor is it simple to reconcile our results with canonical denitrification coupled to nitrification for the reasons described above.

I have few other comments, provided in an annotated copy.

Reviewer 3's comments from their annotated copy.

Abstract, line 23. Neither the pathway nor the type of coupling have to be new. Maybe just less present in the minds of most researchers.

As detailed above.

The abstract provides very little concrete information at the moment and is therefore difficult to understand for readers who have not read the whole text.

As for reviewer 2, we have revised the abstract throughout and now include a key "simple" fact identified through this revision. We do however want to keep any technical jargon to a minimum for a more general audience.

Line 38. Although bacterial denitrification is mostly anaerobic, aerobic denitrification has also been discussed in the literature.

There is a lot of potential confusion in the literature when comes to the terminology used to describe oxygen. We prefer to stick to the convention of using oxic and anoxic to describe the presence or absence of measurable oxygen in an environment and aerobic and anaerobic when stating the mode of respiration on either oxygen or alternative electron acceptors e.g. NO_3^- , NO_2^- or SO_4^{2-} etc., respectively. Yes, denitrification in the presence of

oxygen i.e., in an oxic environment has been recorded and discussed but that denitrification is still anaerobic because – by definition – it is using nitrate or nitrite as alternative electron acceptors, which is what we refer to at this point in the text. Lines 36-41.

Line 62. This may be true in aquatic systems. In soils, 'free mixing' is inherently difficult, although it might be easier for nitrite and nitrate than e.g. for ammonia. Anyway, we also know that nitrifiers, for example, can reduce nitrite further to N₂O and also N₂ in nitrifier denitrification. Although they can take up nitrite from the surrounding, they will also produce nitrite and directly reduce it further.

With regard to free mixing. We have added “sediments” to indicate that we are focusing on aquatic systems, as a full description of the problems with mixing ammonia in non-saturated soils, or soils in general, is beyond the scope of our manuscript even though they are a major global source of N₂. See lines 57-64.

With regard to the second point. Yes, for example, Shaw et al. (2006) did demonstrate that pure cultures of Nitrosomonas europaea could denitrify both their own endogenous nitrite along with exogenous nitrite to N₂O (in effect our treatments 1, 3 & 5), yet we saw no consistent production of ¹⁵N-N₂ from ¹⁵N-nitrite and no difference in ²⁹N₂ production from ¹⁵N-ammonia with or without ¹⁴N-nitrite. As above, see lines 131-134 and 183-197. This is now our major reason for not supporting nitrifier-denitrification or canonical denitrification as our source of N₂ gas from ¹⁵N-ammonia. We do not wish to get into an argument over complete nitrifier-denitrification to N₂ which is why we emailed the reviewer (as they had signed their review) for clarification on this point.

*The reviewer's claim above that “we also know that nitrifiers... reduce nitrite to N₂O and **also N₂** in nitrifier denitrification” is not actually known for sure for N₂. Instead and as we state, complete nitrifier-denitrification to N₂ would require nitrifiers to have nitrosocyanin as a plausible substitute to canonical N₂O-reductase to enable complete reduction of nitrite to N₂ gas¹¹.*

Line 70. This is not a complete sentence, please rephrase.

This whole section has been revised, see lines 67-70.

Line 123. In studies of nitrifier denitrification, authors distinguish so-called 'fertiliser denitrification' from 'nitrification-coupled denitrification', assuming the use of different substrate pools (see e.g. Kool et al., 2011, Methods in Enzymology 496).

The original isotope pairing technique developed by Nielsen in 1992¹⁵ was designed to do just that in whole, intact sediment cores and he coined the terms D_w and D_n. Where D_w is denitrification of “fertiliser” nitrate originating from the overlying water in the sediment core and D_n, denitrification of nitrate borne from nitrification in the upper oxic sediment layers i.e. coupled nitrification-denitrification. Even with the advent of anammox in 2002 this distinction could still be made i.e. Trimmer et al., 2006¹⁶. However, for the technique to work properly, the added ¹⁵NO₃⁻ must be allowed to mix by diffusion with ¹⁴NO₃⁻ already in the system, which can be checked by plotting the ratio of ²⁹N₂ to ³⁰N₂ production over time¹³. Even though we can distinguish between these two sources of nitrate, there is still only one pool – fed from the two sources - at the point of “denitrification”. For example, if we imagine a system with no ¹⁴NO₃⁻ in the overlying water but a strong nitrification, ¹⁴NO₃⁻ producing potential in the oxic sediments below – the North Sea in May is analogous. We add ¹⁵NO₃⁻ to the overlying water which diffuses in and mixes with ¹⁴NO₃⁻ borne in the sediment. Denitrification will then produce ³⁰N₂ for D_w and ²⁹N₂ for D_n but there is only one mixed pool (F_N) of ¹⁴NO₃⁻ and ¹⁵NO₃⁻ integrated in the production of ²⁹N₂ and ³⁰N₂. What we are reporting

now for oxic, permeable sediments is distinct. Here we added $^{15}\text{NH}_4^+$ that we know was oxidised to produce $^{15}\text{NO}_3^-$ and $^{15}\text{NO}_2^-$ in the porewater ($^{15}\text{NO}_x^-$, new Fig. 1b), however, that $^{15}\text{NO}_x^-$ appears isolated from the pool of ^{15}N (whatever it may be) being used to make $^{15}\text{N}_2$ gas i.e. there is more than one pool; one measurable in the porewater and one cryptic at the point of “denitrification” (or whatever the process is that produces N_2 gas).

Line 142. Why do you think this is counterintuitively? Ammonia is not directly the substrate for N_2 production. Any effect ammonia oxidation has, will be mirrored in the enrichment of nitrite or nitrate. In natural abundance studies, this might be different, as fractionation plays a larger role and any nitrite measured will only be the 'left-over' and not the real pool seen by the microbes. If the coupling is very tight or N_2 is produced in nitrifier denitrification, the correlation between the measured nitrite pool and the enrichment of N_2 produced will also be minor.

If we add $^{15}\text{NO}_3^-$ to a sediment with a DNRA potential that reduces $^{15}\text{NO}_3^-$ to $^{15}\text{NH}_4^+$ then we will end up with ^{15}N in both the F_N and F_A pools. Song et al. (2016)¹⁰ proposed a formulation to handle DNRA, denitrification and anammox which includes ^{15}N in the F_A , F_N and F_{N_2} pools and it is that formulation that people who work with sediments are familiar with. NOTE – counter to the comment “Ammonia is not directly the substrate for N_2 production” with anammox it is (reaction 4). That is why, from the point of view of Song, and intuitively to many others, both F_A and F_N must influence F_{N_2} (but only if all substrates mix which might not be true for soils!). Hence, if we now show that F_A is redundant then that will be counter intuitive to people familiar with that accepted formulation. We have, however, rephrased the text see lines 141-156. The point about any nitrite measured in the porewater will only be the 'left-over' and not the real pool seen by the microbes is a very good point that we now include on lines 171-172.

Line 228. Also same ratio of air to slurry? They were shaken, correct?

Yes, the same ratios were used throughout all of the incubations described and yes they were all mixed by gentle rotation. See lines 236 and 244.

Line 239. As far as I understand, you measured in the normal background of 80% N_2 in the overhead. How did this work?

Do you mean how do we detect ^{15}N atom % enrichment in N_2 above natural abundance? The mass-spec that we use for this work has a sensitivity of about 0.1‰ N which, in the dimensions described here, translates to $\sim 0.08 \text{ nmol } ^{29}\text{N}_2 \text{ g}^{-1}$ dry sed. In Fig. 1, total ^{15}N - N_2 had already accumulated to about $2 (\pm 0.3) \text{ nmol N g}^{-1}$ dry sed after 1 hour which is easily in excess of the detection limit and we have done it similarly since Trimmer first started reporting anammox in estuarine sediments back in 2003 of $1\text{-}10 \text{ nmol N g}^{-1}$ wet sediment h^{-1} (back then). Admittedly, the mass spec is less sensitive to $^{30}\text{N}_2$, so to calculate the ratios we only use the data $>1 < 10\text{h}$ when $^{30}\text{N}_2$ is typically above $0.1 \text{ nmol N g}^{-1}$ dry sed. There are those who advocate degassing water with He first to lower the background for more sensitive work in OMZ waters but that is not the case here – but even that is tricky because you need to make sure you lower the N_2 to same concentration in each vial.

Line 292. Looks like it is up to maybe 11 h.

Thanks for pointing this out. The last time point with data before the plateau is actually 9h, so we have recalculated the ratios for the data $>0 < 10$. The simple overall average rates of production in Fig. 1a and Table 1 are for the overall 12 hours.

Supplementary equations. ...denitrification, D,...

Done, along with a complete revision of the equations throughout the main text and supplement.

Nicole Wrage-Mönnig

References

- 1 Hall, G. H. Measurement of nitrification rates in lake-sediments - comparison of the nitrification inhibitors nitrapyrin and allylthiourea. *Microb. Ecol.* **10**, 25-36 (1984).
- 2 Lansdown, K. *et al.* Importance and controls of anaerobic ammonium oxidation influenced by riverbed geology. *Nature Geosci* **9**, 357-360 (2016).
- 3 Jensen, M. M., Thamdrup, B. & Dalsgaard, T. Effects of specific inhibitors on anammox and denitrification in marine sediments. *Appl. Environ. Microbiol.* **73**, 3151-3158 (2007).
- 4 Klawonn, I. *et al.* Distinct nitrogen cycling and steep chemical gradients in *Trichodesmium* colonies. *ISME J.* **14**, 399-412 (2020).
- 5 Shen, L., Ouyang, L., Zhu, Y. & Trimmer, M. Spatial separation of anaerobic ammonium oxidation and nitrite-dependent anaerobic methane oxidation in permeable riverbeds. *Environ. Microbiol.* **21**, 1185-1195 (2019).
- 6 Shelley, F., Grey, J. & Trimmer, M. Widespread methanotrophic primary production in lowland chalk rivers. *Proc. Roy. Soc. London Ser. B* **281** (2014).
- 7 Broda, E. 2 kinds of lithotrophs missing in nature. *Z. Allg. Mikrobiol.* **17**, 491-493 (1977).
- 8 Richards, F. A., Cline, J. D., Broenkow, W. W. & Atkinson, L. P. Some consequences of the decomposition of organic matter in Lake Nitinat, an anoxic fjord. *Limnol. Oceanogr.* **10**, 185-201 (1965).
- 9 van de Graaf, A. A. *et al.* Anaerobic oxidation of ammonium is a biologically mediated process. *Appl. Environ. Microbiol.* **61**, 1246-1251 (1995).
- 10 Song, G. D., Liu, S. M., Kuypers, M. M. M. & Lavik, G. Application of the isotope pairing technique in sediments where anammox, denitrification, and dissimilatory nitrate reduction to ammonium coexist. *Limnol. Oceanogr. Meth.* **14**, 801-815 (2016).
- 11 Wrage-Mönnig, N. *et al.* The role of nitrifier denitrification in the production of nitrous oxide revisited. *Soil Biol. Biochem.* **123**, A3-A16 (2018).
- 12 Christensen, J. P., Murray, J. W., Devol, A. H. & Codispoti, L. A. Denitrification in continental shelf sediments has major impact on the oceanic nitrogen budget. *Global Biogeochem. Cy.* **1**, 97-116 (1987).
- 13 Trimmer, M. & Nicholls, J. C. Production of nitrogen gas via anammox and denitrification in intact sediment cores along a continental shelf to slope transect in the North Atlantic. *Limnol. Oceanogr.* **54**, 577-589 (2009).
- 14 Wang, S. Y. *et al.* Anaerobic ammonium oxidation is a major N-sink in aquifer systems around the world. *ISME J.* **14**, 151-163 (2020).
- 15 Nielsen, L. P. Denitrification in sediment determined from nitrogen isotope pairing. *FEMS Microbiol. Ecol.* **86**, 357 - 362 (1992).
- 16 Trimmer, M., Risgaard-Petersen, N., Nicholls, J. C. & Engstrom, P. Direct measurement of anaerobic ammonium oxidation (anammox) and denitrification in intact sediment cores. *Marine Ecology-Progress Series* **326**, 37-47 (2006).

REVIEWERS' COMMENTS

Reviewer #1 (Remarks to the Author):

The authors of "Coupled nitrification and N₂ gas production as a cryptic process in oxic riverbeds" have edited the manuscript to make it much clearer. The addition of additional experiments is also appreciated. I find the paper more convincing now, and I think the novelty of the findings is more obvious. I only have minor comments below.

Lines 48-51: I think it is really interesting that Broda also predicted ammonium oxidation to N₂ and I am glad that the authors have added this to the paper. However, I think this mini-paragraph needs one or two more concluding sentences that makes it clear that we have never yet seen this process in nature. Right now it seems strangely connected to the next paragraph. Especially because the first sentence of next paragraph says "these aerobic and anaerobic metabolisms."

Line 58: "the important point to appreciate here is that the products of aerobic nitrification (e.g. nitrate and nitrite) are 'free' to mix with any existing nitrate and nitrite in the surrounding porewater before they are subsequently metabolized"—perhaps it would be better to say "are thought (or assumed) to be free to mix" since you are about to say that this is not true.

Lines 174-182: I still think that the concept that internal N pools derail the accuracy of the isotope pairing technique has been previously published and should be cited here briefly. De Brabandere et al 2014 *Environmental Microbiology* 16: 3041-3054. and Chang et al 2014 *Limnology and Oceanography* 59: 1267-74.

Lines 180-182: It is unclear how your scenario links to CANON wastewater reactors, especially since you don't think anammox is involved. Either explain more or cut.

Line 187-188: Can you elaborate on this slightly?

Reference 20: (Jensen et al 2007) sediments is misspelled.

Reviewer #2 (Remarks to the Author):

This paper is a revision of one that I reviewed before. My first review suggested areas to improve the presentation and flow of this work. I see that the other reviews were also critical yet supportive of the manuscript. This revised version is much clearer than the preceding draft of the manuscript. Indeed I enjoyed this paper much more and now I am thinking hard about N transformations in an oxic aquifer in which I work. I like very much how they use the Broda's thermodynamic theory to set the stage for the paper. I also like the end in which the author's consider the different processes that contribute to the pattern they saw, i.e. coupled transformations vs. a new pathway. I only have a few comments below.

101 must in place of has to

113. Need to state the expectation so readers get the surprise.

125. were too ow. Past tense throughout.

125. This paragraph clearly explains the reasoning behind this paper

166 This?

170 clear now thanks.

230 this was not possible??

Reviewer #3 (Remarks to the Author):

I congratulate the authors on this thorough revision of the manuscript, which is now in my opinion very convincing and well written. I have added an annotated copy of the manuscript with very few editorial remarks and one minor comment.

Detailed reply to final reviewers' comments for Nature Communications manuscript NCOMMS-20-09091A

Reviewer #1 (Remarks to the Author):

The authors of "Coupled nitrification and N₂ gas production as a cryptic process in oxic riverbeds" have edited the manuscript to make it much clearer. The addition of additional experiments is also appreciated. I find the paper more convincing now, and I think the novelty of the findings is more obvious. I only have minor comments below.

Lines 48-51: I think it is really interesting that Broda also predicted ammonium oxidation to N₂ and I am glad that the authors have added this to the paper. However, I think this mini-paragraph needs one or two more concluding sentences that makes it clear that we have never yet seen this process in nature. Right now it seems strangely connected to the next paragraph. Especially because the first sentence of next paragraph says "these aerobic and anaerobic metabolisms."

Ok, appreciated, we have added a brief phrase at the end of line 50 "complete aerobic ammonia oxidation to N₂ gas – that, to the best of our knowledge – has yet to be observed in nature." And slightly rephrased the first line of the following paragraph so that it doesn't appear "strangely connected" to the former. "In estuarine or coastal sea sediments, combinations of recognised aerobic and anaerobic metabolisms (equations 1 to 4) buffer..."

Line 58: "the important point to appreciate here is that the products of aerobic nitrification (e.g. nitrate and nitrite) are 'free' to mix with any existing nitrate and nitrite in the surrounding porewater before they are subsequently metabolized"—perhaps it would be better to say "are thought (or assumed) to be free to mix" since you are about to say that this is not true.

Thanks, good-point, that now echoes the abstract.

Lines 174-182: I still think that the concept that internal N pools derail the accuracy of the isotope pairing technique has been previously published and should be cited here briefly. De Brabandere et al 2014 Environmental Microbiology 16: 3041-3054. and Chang et al 2014 Limnology and Oceanography 59: 1267-74.

Ok, appreciated, have cited the De Brabandere et al. (2014) paper again and now also Trimmer's own original description of this potential (Nicholls, Davies, Trimmer, 2007). See lines 184-189.

Lines 180-182: It is unclear how your scenario links to CANON wastewater reactors, especially since you don't think anammox is involved. Either explain more or cut.

Our scenario does not link to CANON wastewater reactors per se and was never described as such. Our entire paper argues for cryptic, internal N cycling. A CANON reactor combines aerobic oxidation of ammonia to nitrite, on the outside, with anaerobic ammonia oxidation to N₂ - using that very nitrite - on the inside i.e., it is an example of internal N cycling but it is not necessarily our form of internal N cycling. The mechanism in nature beyond a CANON reactor is unknown - be it in the ocean (references above) or our oxic riverbed sediments. See lines 184 to 189.

Line 187-188: Can you elaborate on this slightly?

*There isn't really anything more to elaborate with. We took our steer from the review by Nichole Wrage-Mönnig et al. (2018), reviewer 3 here, who refer to 11 papers when discussing the potential of nitrosocyanin to substitute for canonical N_2O reductase but there is nothing more concrete as to whether it actually produces N_2 from N_2O . We do now cite Arciero et al 2002 who state "Nitrosocyanin (NC), a soluble, red Cu protein isolated from the ammonia-oxidizing autotrophic bacterium *Nitrosomonas europaea*, is shown to be a homooligomer of 12 kDa Cu-containing monomers. Oligonucleotides based on the amino acid sequence of the N-terminus and of the C-terminal tryptic peptide were used to sequence the gene by PCR. The translated protein sequence was significantly homologous with the mononuclear cupredoxins such as plastocyanin, azurin, or rusticyanin, the type 1 copper-binding region of nitrite reductase, and the binuclear CuA binding region of N_2O reductase or cytochrome oxidase." The essence of this was conveyed in our original lines 194 to 195, but we have added "a soluble red Cu protein isolated from *Nitrosomonas europaea*" but do not think it necessary to include any more or, indeed, speculate any further.*

Reference 20: (Jensen et al 2007) sediments is misspelled.

Thanks, and well-spotted. Odd as that is the version downloaded to EndNote – if you look even closer the first 'm' in anammox had also been replaced by the same 'rn'.

Reviewer #2 (Remarks to the Author):

This paper is a revision of one that I reviewed before. My first review suggested areas to improve the presentation and flow of this work. I see that the other reviews were also critical yet supportive of the manuscript. This revised version is much clearer than the preceding draft of the manuscript. Indeed I enjoyed this paper much more and now I am thinking hard about N transformations in an oxic aquifer in which I work. I like very much how they use the Broda's thermodynamic theory to set the stage for the paper. I also like the end in which the author's consider the different processes that contribute to the pattern they saw, i.e. coupled transformations vs. a new pathway. I only have a few comments below.

101 must in place of has to *Done*

113. Need to state the expectation so readers get the surprise.

This is explained in detail in the proceeding lines 101 to 108 but we have rephrased lines 110 to 111 to make the link more explicit – "We tested the validity of this accepted mathematical framework by changing the fraction of porewater NO_x^- labelled with ^{15}N (F_N) and looking for how this influenced the ratio of $^{29}N_2$ to $^{30}N_2$ produced (R). First we directly decreased F_N by adding ^{14}N -nitrite to dilute the ^{15}N -nitrite accumulating in the porewater from the oxidation of ^{15}N -ammonia (Treatments 1 to 4, Table 1). Surprisingly,..."

125. were too ow. Past tense throughout. *Done*.

125. This paragraph clearly explains the reasoning behind this paper. *Thanks*.

166 This? *If you meant replace "here" with "this", then we have done that.*

170 clear now thanks. *Good, thanks*.

230 this was not possible?? *Done, thanks.*

Reviewer #3 (Remarks to the Author):

I congratulate the authors on this thorough revision of the manuscript, which is now in my opinion very convincing and well written. I have added an annotated copy of the manuscript with very few editorial remarks and one minor comment.

Reviewer 3's comments from their annotated pdf

Line 56. Change (1-10-100kms) to (1-100km). *Done.*

Line 91-93. I suggest rewriting: ...as $^{15}\text{NO}_x$, i.e. as either ^{15}N -nitrite (equation 1) or the final... (equation 2).

We have inserted an "i.e." as indicated. That is "...as $^{15}\text{NO}_x^-$, i.e. as either..."

Line 168. This clause appears incomplete.

*It would be clearer to highlight the clause in question and attach the comment directly to that "pop-up" box but, what we have is: "In contrast, if we again force denitrification to be the only source of N_2 , and calculate FN_2 assuming that $\text{FN} = \text{FN}_{\text{pw}}$ (Fig. 2a), then the points fall away from our measured R values". Do you mean something like "In contrast, if we again force denitrification to be the only source of N_2 , and calculate FN_2 **by** assuming that **FN_{pw} is equal to FN** (Fig. 2a), then the **resulting data** points fall away from our measured R values". We are not convinced that this adds very much and will wait for the editor's decision, ok?*

Line 191-192. Well, Liz Shaw and co-workers showed that up to 13.5% of N_2O produced by pure culture nitrifiers was from exogeneous nitrite. The rest was not labelled when labelled nitrite was offered - not ruling out nitrifier denitrification of intrinsic nitrite as a source of this unlabelled N_2O , though. Just a comment.

Ok, the reference we cite at the end of line 191 is indeed the Liz Shaw paper and that is the very potential that we are recognizing in the presence of exogenous $^{15}\text{NO}_2^-$ - albeit for the production of N_2 and not N_2O but we don't see any consistent production of N_2 like we do when we add $^{15}\text{NH}_4^+$. But as this is only a comment we will not pursue it any further.

Line 230. Was...

Replaced "what" with "was". Thanks for spotting that.

Line 257. Please insert a space. *Done for 0 h, as for the others.*

Line 278. Remove brackets. *Done.*

Reference 37. Seems the same as reference # 18. *Thanks, an error in Endnote, corrected.*